# GENERALIZED PREF-SHAP TO EXPLAIN PREFERENCE FUNCTIONS

## ABSTRACT

We address the problem of feature attribution for skew-symmetric preference functions in dueling data settings, using the cooperative game-theoretic concept of *Shapley values*. Building on Pref-SHAP[Hu et al. (2022)], we propose *Generalized Pref-SHAP*, a framework that extends its applicability to a broader class of preference functions. Our method leverages a simple neural network to model arbitrary feature mappings while exploiting the canonical block structure inherent to skew-symmetric functions, enabling more meaningful explanations. Additionally, we explore foundational questions about Pref-SHAP, including its relationship with the block decomposition structure of skew-symmetric generalized preference function (GPM)[Hu et al. (2022)]. We perform experiments on a range of synthetic datasets to demonstrate the effectiveness and efficiency of our approach.

## 1 INTRODUCTION

Pairwise preference learning, often modeled via *dueling data*, plays a central role in ranking, recommender systems, sports tournaments, voting, online games etc. A common framework for modeling such pairwise data involves *skew-symmetric preference functions*, i.e., the function $f(u,v)$ satisfies $f(u,v) = -f(v,u)$, representing the preference for item $u$ over item $v$, where $u,v \in \mathbb{R}^d$. We can represent such functions using the following *canonical form*[Rajkumar et al. (2021),Veerathu & Rajkumar (2021)]:

$$f(u,v) = u^\top A v, \tag{1}$$

where, $d$ is even and $A \in \mathbb{R}^{d \times d}$ is a block-diagonal matrix composed of $d/2$ skew-symmetric $2 \times 2$ rotation matrices $\begin{bmatrix} 0 & -1 \\ 1 & 0 \end{bmatrix}$. This bilinear structure naturally arises in modeling utilities and comparisons, and is particularly amenable to analysis due to its algebraic simplicity and geometric structure. While modeling preference functions has received considerable attention[Negahban et al. (2015),Rajkumar & Agarwal (2016),Chen & Joachims (2016),Makhijani & Ugander (2019),Bower & Balzano (2020)], explaining their predictions, that is, attributing the model's output to individual input features, remains less explored. Recent work, notably Pref-SHAP[Hu et al. (2022)], extends the classical Shapley value[Shapley et al. (1953)] to the setting of pairwise comparisons, assigning feature-level attributions to preference decisions using a game-theoretic lens. We aim to investigate both the theoretical and practical behaviors of Pref-SHAP in non-parametric model setting based on the canonical form.

### 1.1 MAIN CONTRIBUTIONS

- **Block Structure Consistency:** We analyze whether Pref-SHAP respects the block structure inherent in the canonical form. This reveals how well the Shapley attributions align with the inherent feature pairing in the model.

- **Theoretical Analysis Under Canonical Form:** We derive closed-form expressions for Pref-SHAP values when $f(u,v) = u^\top A v$, focusing on the two-feature case for clarity. We explore how the attributions are affected by distributional properties like feature variance, constancy, and independence. Our analysis identifies unintuitive behaviors that arise from interaction effects and the symmetry axiom in Shapley values, and suggests that interaction-aware explanations may be necessary in some cases.

- **Generalization to Arbitrary Feature Mappings[2]:** We extend Pref-SHAP to work with nonlinear or learned feature maps $\phi$, proposing *Generalized Pref-SHAP*. This new framework preserves the interpretability of Shapley decompositions while leveraging the structure in $A$, enabling explanations in complex models with deep representations.

## 2 BACKGROUND MATERIALS

### 2.1 PREFERENCE LEARNING

We consider a more general class of skew-symmetric functions of the form:

$$f(u, v) = \langle \phi(u), A\phi(v) \rangle, \tag{2}$$

where $u, v \in \mathbb{R}^k$, $\phi : \mathbb{R}^k \to \mathbb{R}^d$ is a feature mapping(possibly nonlinear) with even $d$, and $A \in \mathbb{R}^{d \times d}$. Any skew-symmetric function can be represented in this form, where $\phi$ is the identity function and $k = d$. The likelihood function used for such models, especially in pairwise preference scenarios, often takes the form:

$$p(y \mid u, v) = \sigma(y \cdot f(u, v)) = 1 - p(y \mid v, u), \tag{3}$$

where $\sigma(x) = \frac{1}{1+e^{-x}}$ is the sigmoid function and $y \in \{-1, 1\}$ indicates the preference label.

### 2.2 SHAPLEY VALUE

The purpose of this paper is to study the local explainability of predictions based on preference functions using the *Shapley value*[Shapley et al. (1953)], a concept from cooperative game theory. It is basically a credit allocation method where a subset of $k$ players is assigned a value function based on their contribution to a game $\nu : [0, 1]^k \to \mathbb{R}$. The Shapley value for player $j$ in game $\nu$ is given by:

$$\Phi_j(\nu) = \sum_{S \subseteq \Omega \setminus \{j\}} \frac{|S|!(d - |S| - 1)!}{d!} \left[ \nu(S \cup \{j\}) - \nu(S) \right],$$

where, $\Omega = \{1, \ldots, d\}$ is the set of $d$ players.

It satisfies several desirable uniqueness axioms such as efficiency(6), symmetry, null player property, linearity. The use of this concept in explainable ranking context helps in finding feature attribution in a model prediction by creating a correspondence between the concept of players and item features. In particular, linearity ensures that for a linear ensemble of models, the Shapley value of a feature is the corresponding linear combination of individual model Shapley values.

### 2.3 PREF-SHAP

In the Pref-SHAP framework[Hu et al. (2022)], the value function is adapted to model the pairwise preference setting.

### 2.4 PREFERENTIAL VALUE FUNCTION FOR ITEMS[HU ET AL. (2022); CHAU ET AL. (2022B), GRÜNEWÄLDER ET AL. (2012)]

**Definition 1.** *Given a preference function $f \in \mathcal{H}$, and a pair of items $(x^l, x^r) \in \mathcal{X} \times \mathcal{X}, \mathcal{X} \subseteq \mathcal{R}^d$, the preferential value function $\nu : \mathcal{X} \times \mathcal{X} \times [0, 1]^k \times \mathcal{H} \to \mathbb{R}$ for computing Shapley values ($\Phi$) in Pref-SHAP is defined as the following conditional expectation:*

$$\nu_{x^l, x^r, S}(f) = \mathbb{E}_r \left[ f(\{X_S^l, X_{S^c}^l\}, \{X_S^r, X_{S^c}^r\}) \mid X_S^l = x_S^l, \ X_S^r = x_S^r \right], \tag{4}$$

*where $S \subseteq \{1, \ldots, k\}$ is a subset of feature indices, and $X$ is the input random vector. The notation $X_S$ denotes the subvector of features indexed by $S$, and $S^c$ is the complement of $S$. The pair $\{X_S, X_{S^c}\}$ refers to the full input vector $X$ formed by concatenation. The reference distribution $r$ is defined as:*

$$r = r \left( X_{S^c}^l, X_{S^c}^r \mid X_S^l = x_S^l, X_S^r = x_S^r \right).$$

## 2.5 GENERALIZED PREFERENTIAL KERNEL[CHAU ET AL. (2022A)]

**Definition 2.** *Given a kernel $k : \mathcal{X} \times \mathcal{X} \to \mathbb{R}$, defined on the original feature space $\mathcal{X} \subseteq \mathcal{R}^d$, the Generalized Preferential Kernel $k_E$ is defined as:*

$$k_E\big((x_i^l, x_i^r), (x_j^l, x_j^r)\big) = k(x_i^l, x_j^l) \cdot k(x_i^r, x_j^r) - k(x_i^l, x_j^r) \cdot k(x_i^r, x_j^l), \tag{5}$$

*where the skew-symmetric function $f$ is assumed to lie in the Reproducing Kernel Hilbert Space (RKHS) $\mathcal{H}_{k_E}$ associated with $k_E$.*

## 2.6 BLOCK STRUCTURE OF SHAPLEY VALUES

One of the key properties satisfied by Shapley values, and more importantly for our purposes by Pref-Shap, is the *efficiency* axiom which states that the sum of all feature attributions equals the overall preference score. In particular, for a feature space of dimension $d$ where features are grouped into $d/2$ disjoint consecutive blocks of size 2, we have the following equation w.r.t. Pref-SHAP ($\Phi$)(8):

$$\sum_{i=1}^{d/2} (\Phi_{2i-1} + \Phi_{2i}) = \sum_{i=1}^{d/2} (u_{2i-1}v_{2i} - u_{2i}v_{2i-1}) = u^\top A v, \tag{6}$$

where, $u, v \in \mathbb{R}^d$, $d$ is even, and $A \in \mathbb{R}^{d \times d}$ is a block-diagonal matrix composed of $d/2$ skew-symmetric $2 \times 2$ rotation matrices $\begin{bmatrix} 0 & -1 \\ 1 & 0 \end{bmatrix}$ as defined in (1). Each block corresponds to an anti-symmetric interaction between two consecutive features. A natural question that arises is whether Pref-SHAP also satisfies a finer-grained *block decomposition* property at the level of individual 2-dimensional feature blocks. That is, for each odd index $i \in \{1, 3, \ldots, d-1\}$, does the following hold?

$$\Phi_i + \Phi_{i+1} = u_i v_{i+1} - u_{i+1} v_i = u^\top A_{i:i+1} v, \text{ for } i \bmod 2 \neq 0, \tag{7}$$

where $A_{i:i+1} \in \mathbb{R}^{d \times d}$ is a matrix whose only nonzero entries lie in a $2 \times 2$ skew-symmetric submatrix spanning rows and columns $i$ and $i+1$, and zeros elsewhere. Thus, the full matrix $A$ can be written as a sum of these block-local matrices: $A = \sum_{i=1}^{d/2} A_{2i-1:2i}$. Here, $A_{2i-1:2i}$ denotes the $2 \times 2$ submatrix of $A$ corresponding to rows and columns $2i - 1$ and $2i$. This structure leads us to ask: under what conditions does Pref-SHAP decompose additively over such blocks, preserving the local attribution property in Eq. equation 7?

## 3 DOES PREF-SHAP OBEY THE BLOCK PATTERN?

**Proposition 1** (Block Decomposition of Conditional Pref-SHAP under Independence). *Consider a skew-symmetric preference function $f : \mathcal{X} \times \mathcal{X} \to \mathbb{R}$ (Definition 1) defined on feature vectors $u, v \in \mathbb{R}^d$, where the $d$ features are partitioned into $d/2$ disjoint consecutive blocks: $B_j = \{2j - 1, 2j\}, \quad j = 1, \ldots, \frac{d}{2}$. Assume one of the following:*

- ***Full independence:** All features $\{X_1, \ldots, X_d\}$ are mutually independent, or*

- ***Blockwise independence:** Features within each block $B_j$ may be dependent, but blocks are mutually independent, i.e., $X_{B_i} \perp X_{B_j}$ for all $i \neq j$, where $X_{B_j}$ denotes the features in block $B_j$.*

*The preference function decomposes additively over blocks: $f(u, v) = \sum_{j=1}^{d/2} f_j(u_{B_j}, v_{B_j})$, where each $f_j$ depends only on features in block $B_j$.*

*Pref-SHAP for feature $i \in [d]$ is analytically computed as*

$$\Phi_i := \sum_{S \subseteq [d] \setminus \{i\}} \frac{|S|!(d - 1 - |S|)!}{d!} \big[\nu(S \cup \{i\}) - \nu(S)\big], \tag{8}$$

*where the conditional value function is, $\nu(S) := \mathbb{E}\big[f(u, v) \mid (u_k, v_k)_{k \in S}\big]$.*

*Then, under the above assumptions, $\sum_{i \in B_j} \Phi_i = f_j(u_{B_j}, v_{B_j})$,   for all $j = 1, \ldots, \frac{d}{2}$. Moreover, the individual Pref-SHAP values $\Phi_i$ generally depend on the full feature set due to the global conditioning in $\nu$, but their sum over each block recovers the exact blockwise preference contribution. If the features are correlated across blocks (i.e., the blockwise independence assumption fails), then this additive decomposition of the Pref-SHAP values does not generally hold.*

**Proof Sketch.** Consider features partitioned into blocks $B_1, \ldots, B_m$. Pref-SHAP values are computed via differences of conditional expectations. Under full or blockwise independence, conditional expectations of features outside a block reduce to constants independent of conditioning subsets. Consequently, cross-block terms appear symmetrically in the Shapley difference terms and cancel out. This implies that Pref-SHAP values decompose additively over blocks, with each block's attribution depending only on its own features. In contrast, when features are arbitrarily correlated across blocks, these cancellations no longer occur, and the decomposition fails. The complete proof is provided in the appendix B.

## 3.1 CAN OTHER SHAPLEY VARIANTS RECOVER BLOCK PATTERN(7)?

A natural question is whether using other value functions can help Pref-SHAP respect the canonical block structure in cases where the original conditional value function leads to violation. Specifically:

**Off-manifold / Interventional / Marginal Shapley values**[Janzing et al. (2020)] evaluate features outside the data distribution, which may help remove interdependencies across blocks. However, such approaches lack robustness[Slack et al. (2020)] because they evaluate the model on unrealistic samples, potentially making the explanations unreliable and vulnerable to adversarial manipulation.

**ManifoldSHAP**[Taufiq et al. (2023)] attempts to stay on the data manifold by estimating it via kernel density estimators or score models. Since Pref-SHAP involves estimating conditional expectations from a distributional perspective, Kernel Mean Embeddings could serve as a tool for implementing ManifoldSHAP if applied in the Pref-SHAP context.

**Causal Shapley values**[Heskes et al. (2020)] incorporate the underlying causal graph and account for structural dependencies between features. This allows feature attribution to reflect true causal contributions, thereby improving interpretability over interventional or marginal approaches in the presence of feature correlation.

As an illustration, consider the Gaussian setup:
$X = [X_a, \ X_b]^T = [X_2, \ X_4, \ X_1, \ X_3]^T$,   where $X_a = \{X_2, X_4\}$, and $X_b = \{X_1, X_3\}$.
Assume $X \sim \mathcal{N}(\mu, \Sigma)$. The conditional expectation and covariance of $X_a$ given $X_b$ are as follows:
$\mathbb{E}[X_a | X_b] = \mu_a + \Sigma_{ab}\Sigma_{bb}^{-1}(x_b - \mu_b), \ \ \Sigma_{a|b} = \Sigma_{aa} - \Sigma_{ab}\Sigma_{bb}^{-1}\Sigma_{ba}$.

Notably, the conditional mean does not depend on $\Sigma_{aa}$. This implies that even if features within a block (e.g., $X_a$) are highly correlated, their internal dependency does not affect the conditional expectation as long as $X_b$ is fixed. However, dependency *across* blocks will break the block decomposition property under the conditional value function.

Hence, while conditional value functions capture statistical dependence, they may cause Pref-SHAP to violate canonical structure, leading to unintuitive or biased attributions, especially when sensitive features are involved. In such cases, alternative value functions like causal or manifold-based ones may provide more reliable and fair explanations.

## 3.2 CAN WE LEARN A FEATURE MAPPING TO RESTORE BLOCK STRUCTURE(7)?

We investigate whether it is possible to map the original features to a transformed space, such as an *eigenspace defined by the covariance matrix* so that the Pref-SHAP values exhibit a block structure in the transformed coordinates. Specifically, we consider an orthonormal linear transformation $W \in \mathbb{R}^{d \times d}$ such that $x = W^T u, \ \ y = W^T v$, where $u, v \in \mathbb{R}^d$ are original feature vectors, and $x, y \in \mathbb{R}^d$ are transformed features. $W = \begin{bmatrix} w_{11} & w_{21} & \cdots & w_{d1} \\ w_{12} & w_{22} & \cdots & w_{d2} \\ \vdots & \vdots & \ddots & \vdots \\ w_{1d} & w_{2d} & \cdots & w_{dd} \end{bmatrix}, u = Wx, \ \ v = Wy$ are the inverse

transformations .

Each transformed feature is, $\quad x_i = \sum_{j=1}^{d} w_{ji} u_j, \quad y_i = \sum_{j=1}^{d} w_{ji} v_j, \quad \forall i = 1, \ldots, d.$

**Example: Skew-Symmetric Function with 4 Features**  Consider the skew-symmetric preference function $f(u, v) = u_1 v_2 - u_2 v_1 + u_3 v_4 - u_4 v_3$, which exhibits a natural block structure with two blocks: $\{1, 2\}$ and $\{3, 4\}$.

Expressing $f$ in the transformed space,

$$f(u, v) = f(Wx, Wy) = (Wx)_1(Wy)_2 - (Wx)_2(Wy)_1 + (Wx)_3(Wy)_4 - (Wx)_4(Wy)_3$$

$$= \sum_{i,j=1}^{d} (w_{i1}w_{j2} - w_{i2}w_{j1} + w_{i3}w_{j4} - w_{i4}w_{j3})(x_i y_j - x_j y_i).$$

Rearranging terms into differences of products (to maintain skew-symmetry), $f(x, y) = \sum_{i<j} c_{ij}(x_i y_j - x_j y_i)$, where, $c_{ij} = w_{i1}w_{j2} - w_{i2}w_{j1} + w_{i3}w_{j4} - w_{i4}w_{j3}$.

Due to the linear mixing by $W$, the original block structure (only interactions within blocks $\{1, 2\}$ and $\{3, 4\}$) generally disappears. Instead, $f$ becomes a *fully connected* skew-symmetric form involving all pairs $(i, j)$ of transformed features, where all $c_{ij}$ can be nonzero. Hence, the model in the transformed space no longer decomposes into independent blocks. Consequently, the Pref-SHAP values computed on transformed features will not obey the block pattern seen in the original features. This observation generalizes to higher dimensions and nonlinear models with interaction terms: the orthonormal transform $W$ mixes all features, distributing interactions across transformed features. Nonlinear terms (e.g., products of features from different blocks) are generally not diagonalized by $W$. Therefore, block independence is a property of the function structure, not solely the feature covariance. Shapley values are generally not linear in the features or their transformations, except for models based on a single item features. For such functions that act on a single item, $f(u) = \beta^T u$, transforming features linearly as $x = W^T u$ yields $f(u) = \beta^T u = \beta^T W x = \tilde{\beta}^T x$, and Shapley values transform linearly: $\phi_{u_j} = \sum_i w_{ji} \phi_{x_i}$. For *pairwise preference* models with feature interactions, this linearity breaks, and Shapley values of original features cannot be represented as linear combinations of transformed features' Shapley values.

**Remark:** Mapping features to an eigenspace or another orthonormal basis does not, in general, preserve the block independence structure of nonlinear interaction models like skew-symmetric preference functions. Consequently, Pref-SHAP values computed in the transformed space will reflect fully coupled feature interactions and fail to obey the original block pattern.

## 4  STUDY OF PREF-SHAP PROPERTIES IN THE TWO-FEATURES SETTING

When there are only two features, the analytical expression for Pref-SHAP(8) reduces to:

$$\begin{aligned}
\Phi_1 = \frac{1}{2}[&(u_1 v_2 - u_2 v_1) + u_1 \mathbb{E}[Y_2|Y_1 = v_1] - v_1 \mathbb{E}[X_2|X_1 = u_1] \\
&- v_2 \mathbb{E}[X_1|X_2 = u_2] + u_2 \mathbb{E}[Y_1|Y_2 = v_2] - \mathbb{E}[X_1]\mathbb{E}[Y_2] + \mathbb{E}[Y_1]\mathbb{E}[X_2]] \\
\Phi_2 = \frac{1}{2}[&(u_1 v_2 - u_2 v_1) - u_1 \mathbb{E}[Y_2|Y_1 = v_1] + v_1 \mathbb{E}[X_2|X_1 = u_1] \\
&+ v_2 \mathbb{E}[X_1|X_2 = u_2] - u_2 \mathbb{E}[Y_1|Y_2 = v_2] - \mathbb{E}[X_1]\mathbb{E}[Y_2] + \mathbb{E}[Y_1]\mathbb{E}[X_2]]
\end{aligned} \tag{9}$$

Since $\mathbb{E}(X_i) = \mathbb{E}(Y_i) \ \forall i \in \{1, 2\}$, the last two terms vanish.

### 4.1  IMPACT OF FEATURE VARIANCE ON PREF-SHAP

Let us consider this analytically. Suppose the features are independent, then the Pref-SHAP values simplify to:

$$\begin{aligned}
\Phi_1 = \tfrac{1}{2}\big[&(u_1 v_2 - u_2 v_1) + (u_1 \mathbb{E}(Y_2)) - (v_1 \mathbb{E}(X_2)) - (v_2 \mathbb{E}(X_1)) + (u_2 \mathbb{E}(Y_1))\big] \\
\Phi_2 = \tfrac{1}{2}\big[&(u_1 v_2 - u_2 v_1) - (u_1 \mathbb{E}(Y_2)) + (v_1 \mathbb{E}(X_2)) + (v_2 \mathbb{E}(X_1)) - (u_2 \mathbb{E}(Y_1))\big]
\end{aligned} \tag{10}$$

Now, suppose, $\text{Var}(X_1) = \text{Var}(Y_1) = 100$ and $\text{Var}(X_2) = \text{Var}(Y_2) = 1$. Consider an instance where $u_1 = 100$, $v_1 = -100$, $u_2 = 1$, $v_2 = -1$. In this case, the first-order terms involving $u_1$ and $v_1$ dominate, so $\Phi_1$ could be significantly larger than $\Phi_2$. However, for another instance with opposite signs (e.g., $u_1 = -100$, $v_1 = 100$, $u_2 = -1$, $v_2 = 1$), $\Phi_2$ could become larger than $\Phi_1$.

Therefore, while individual Pref-SHAP values may be affected by the variance in the features, this effect can cancel out when averaging over many samples. In other words, the global (dataset-level) Pref-SHAP attributions are not directly biased by feature variance in expectation under independence. Hence, Pref-SHAP with conditional value functions represents the true conditional expectation of the model output. Thus, it does not directly encode feature variance. However, the uncertainty in prediction caused by high feature variance (especially in correlated settings) may influence the attribution in an indirect way. This distinction aligns with insights from information-theoretic approaches to attribution [Watson et al. (2024)], where variance influences model uncertainty but not necessarily marginal attributions unless the model or attribution method explicitly encodes that dependency.

## 4.2 EFFECT OF CONSTANT FEATURES ON PREF-SHAP

The full discussion with analysis is deferred to the appendix C due to space constraints. Briefly, even if a feature is constant across item pairs, it may still receive non-zero attribution under Pref-SHAP feature correlations. However, under certain independence assumptions and symmetry axiom, such features may yield equal attributions due to model structure.

| Case | Baseline Form | $\Phi_1$(9) |
|------|---------------|-------------|
| 1 | $z = \frac{u+v}{2}$ (Pair-specific baseline) | $\frac{1}{2}(u_1 v_2 - u_2 v_1 + u_1 u_2 - v_1 v_2)$ |
| 2 | $z = \mathbb{E}[Z]$ (Global baseline) | $\frac{1}{2}(u_1 v_2 - u_2 v_1 + (u_1 - v_1)\mathbb{E}[Z_2] - (v_2 - u_2)\mathbb{E}[Z_1])$ |

Figure 1: Comparison of Baseline Shapley Forms

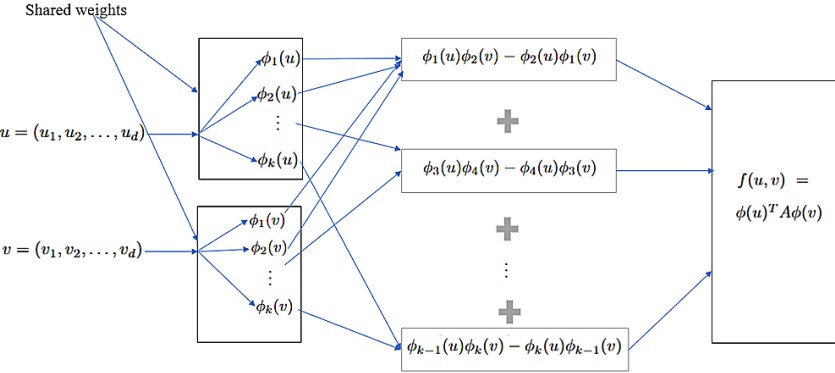

Figure 2: Generalized Pref-SHAP architecture using a simple neural network

## 4.3 PREF-SHAP VS. BASELINE SHAPLEY[SUNDARARAJAN & NAJMI (2020)]

$$\text{Baseline } z = (z_1, z_2) = \left( \frac{u_1 + v_1 + \cdots}{n}, \frac{u_2 + v_2 + \cdots}{n} \right) = \left( \mathbb{E}[Z_1], \mathbb{E}[Z_2] \right) \qquad (11)$$

In Case 1(Fig. 1), when features are equal, attribution vanishes. In Case 2, even equal features may have non-zero attribution if they deviate from the global baseline $\mathbb{E}[Z]$. Pref-SHAP resembles Case 2 and better accounts for meaningful deviations from population averages.

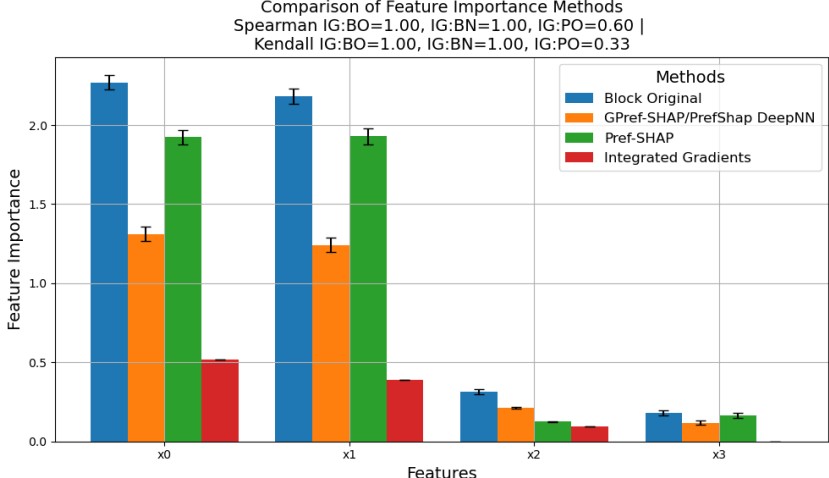

Figure 3: Quadratic feature mappings $\{x_0^2, x_1^2, x_0x_1, x_0x_2\}$. Generalized Pref-SHAP is abbreviated as **GPref-SHAP**; **IG** = Integrated Gradients, **BO** = Block Original, **BN** = Block DeepNN, and **PO** = Pref-SHAP Original. **IG:BO** means the rank correlation between them. The terms **Pref-SHAP DeepNN** and **Block DeepNN** are used interchangeably to refer to GPref-SHAP. Terms in the captions represent the mapped features.

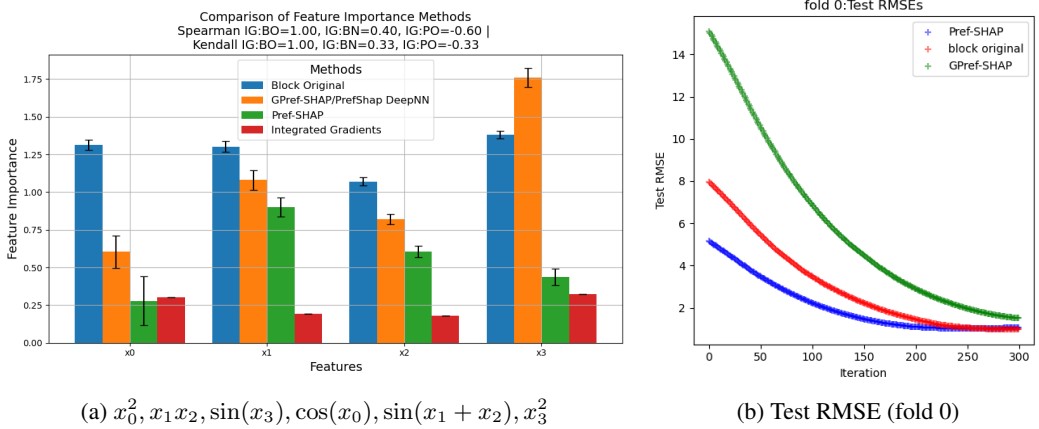

(a) $x_0^2, x_1x_2, \sin(x_3), \cos(x_0), \sin(x_1 + x_2), x_3^2$    (b) Test RMSE (fold 0)

Figure 4: (a)hybrid mapping of features, (b) ReLU network (4 input features, 6 mapped features, 4 hidden layers, 16 nodes in each layer).

## 5 GENERALIZED PREFERENTIAL KERNEL(5) INSIGHT:

Given the bilinear form of the *Generalized Preferential kernel*[Chau et al. (2022a)]:

$$\sum_{i=1}^n \alpha_i K_E((x_i, y_i), (x_{\text{test}}, y_{\text{test}})) = x_{\text{test}}^\top \left( \sum_{i=1}^n \alpha_i (x_i y_i^\top - y_i x_i^\top) \right) y_{\text{test}}. \qquad (12)$$

the model output becomes a bilinear form over input pairs with *weights represented by a skew-symmetric matrix i.e.* $\sum_{i=1}^n \alpha_i (x_i y_i^\top - y_i x_i^\top)$ *in a transformed space*. This naturally aligns with the *linear kernel* as the base kernel (instead of *RBF kernel*), which is both expressive and computationally efficient in this context, i.e. in cases where the feature map $\phi$ is the identity function.

**Effective Feature Space for Pref-SHAP:** In this formulation, the prediction function is linear in the space of effective features, which are formed as pairwise interactions of the original features. As each pair of features $(x_i, x_j)$ contributes an interaction term, the number of effective features becomes $\binom{d}{2}$, significantly expanding the representational capacity. The final prediction is thus: $f(u, v) = \sum_{i<j} \alpha_{ij}(u_i v_j - u_j v_i)$, where $\alpha_{ij}$ are the learned coefficients associated with each

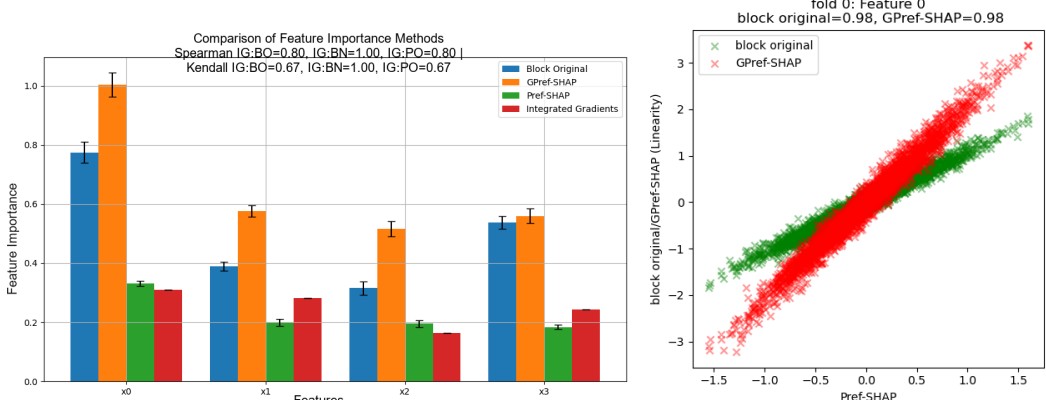

(a) ReLU network (4 input features, 6 mapped features, 4 hidden layers, 16 nodes in each layer)

(b) Linearity of Pref-SHAP variants (fold 0, feature $x_0$)

Figure 5: ReLU network

interaction. This formulation allows us to attribute importance to each original feature based on the strength and frequency of its interactions across data instances.

# 6 PROPOSED FRAMEWORK: GENERALIZED PREF-SHAP (2)

---

**Algorithm 1** Generalized Pref-SHAP

---

**Input:** Pairwise item data $\{X_l, X_r, Y\}$ with features $X \in \mathbb{R}^d$, number of blocks $k$ for feature mapping.

1. Learn the unknown feature mapping $\phi = (\phi_1, \phi_2, \ldots, \phi_k)(2)$ using the neural network shown in Figure 2, where the predicted output is: $f(x_l, x_r) = \phi(x_l)^T A \phi(x_r) = \sum_{i=1}^{k/2} (\phi_{2i-1}(x_l)\phi_{2i}(x_r) - \phi_{2i}(x_l)\phi_{2i-1}(x_r))$.

2. Construct $k/2$ datasets, each corresponding to a blockwise pairwise interaction of mapped features using $\phi$.

3. For each block dataset, apply Kernel Ridge Regression (KRR) to learn a component function.

4. Compute the residual between the original label $Y$ and the sum of predictions from all $k/2$ component models. Apply KRR to this residual dataset.

5. Compute Pref-SHAP values for each of the $k/2$ blocks and the residual component, giving a $k \times (k/2 + 1)$ matrix of attributions.

6. Aggregate feature attributions by summing across columns (i.e., summing attributions across blocks for each feature).

**Output:** Final Pref-SHAP value for each original input feature.

---

In standard Pref-SHAP, the model approximates the skew-symmetric function of the mapped features using Kernel Ridge Regression(KRR) but does not learn the mapping $\phi(2)$ explicitly. Consequently, it cannot exploit the canonical block structure inherent to skew-symmetric functions. In contrast, our proposed *Generalized Pref-SHAP* explicitly learns the feature mapping $\phi(2)$, enabling a structured decomposition of the preference function into interpretable blocks. This decomposition preserves the block structure of the underlying function and facilitates more accurate and meaningful feature attributions. By jointly learning the feature representation and maintaining block-wise interpretability, *Generalized Pref-SHAP* improves the transparency and faithfulness of the attribution process, aligning explanations more closely with the structure of the learned model.

The network takes as input a pair of items represented using their original features and learns the feature mappings of each item via weight sharing. The learned mappings are then passed into a

module which computes the skew-symmetric preference function for these mapped features. The detailed explanation of the algorithmic steps are described in appendix A.

# 7 EXPERIMENTS

In this section, we describe our experimental setup and results on both synthetic and real-world datasets. We have conducted experiments mainly on carefully generated synthetic data because the motivation behind Generalized Pref-SHAP is rooted in the design of feature mappings $\phi(2)$, and real-world datasets rarely provide a ground truth for global feature importance. However, in domains where expert knowledge about feature relevance exists, real-data experiments can help identify the more interpretable model. It is quite possible that the explicit features in the real data get mapped to some hidden space before applying the skew-symmetric preference function instead of directly contributing to the preferences and this phenomenon can be better explained through our proposed method.

## 7.1 SYNTHETIC AND REAL-WORLD EXPERIMENTS

We conduct experiments using synthetic data consisting of $n = 100$ items, where each item's features are sampled from a 4-dimensional Gaussian distribution with mean zero and identity covariance: $x_i \sim \mathcal{N}(0, \mathbf{I})$ for $i = 1, \ldots, 100$. We construct preference labels using various differentiable feature mappings $\phi$, including:
**Polynomial mappings:** quadratic expansions, **Sinusoidal mappings:** combinations of $\sin(\cdot), \cos(\cdot)$ applied to linear projections, **Hybrid mappings:** combinations of polynomial and sinusoidal transformations, **Neural network mappings:** small feedforward networks with ReLU activations and varying depth/width.
The preference labels are generated by applying the skew-symmetric function $f$ to transformed features. We evaluate the feature importance scores computed using four methods: **Pref-SHAP:** the original method using conditional Shapley values on learned KRR models. **Generalized Pref-SHAP (Ours):** learns each skew-symmetric component block via neural networks, then aggregates their Shapley values. **Integrated Gradients (Baseline):** standard attribution baseline for differentiable models. **Block-Original:** a reference method that trains separate KRR models per block and aggregates block-level Pref-SHAP values.

For the ground truth, we compute the global feature importance scores with respect to the feature mapping function using *Integrated Gradients*, and then propagate those scores to the canonical skew-symmetric preference function. Since each block in the canonical form contains exactly two terms, with each feature appearing once per term, the importance score of each feature is doubled per block. As the blocks are disjoint, the final feature importance is obtained by summing these contributions across blocks. This enables comparison using rank-based metrics. We use all $\binom{100}{2}$ unique pairs for training and evaluation, as the task is regression-based whereas our real-world experiment is based on classification of pairwise preferences. The pairwise comparisons with labels are split into training ($80\%$), validation ($10\%$), and test ($10\%$) sets and the experiments are averaged over 5 folds. Feature attribution is computed on a randomly selected subset of the pairwise data. For Kernel Ridge Regression, hyperparameter tuning is performed via gradient-based optimization using the `Falkon` library. For the neural network setup, we fix two hidden layers and employ Bayesian optimization using `Optuna` with 100 trials to tune the hyperparameters: the number of hidden nodes (shared across layers) is selected from the set $\{32, 64, 96, 128\}$, while the learning rate and weight decay are sampled from logarithmic ranges $[10^{-4}, 10^{-1}]$ and $[10^{-6}, 10^{-2}]$ respectively. We use the mean squared error (MSE) loss and optimize using the Adam optimizer with a batch size of 64 and early stopping based on validation loss. *Real-world* experiment (6) is based on classification, so Kernel Logistic Regression (KLR) is used for modeling the pairwise preferences and the concept of residue modeling is not applicable here. We have used the publicly available dataset *Pokemon*(Nguyen Van Anh (2021)) for the same. The 25 features are mapped into 4 hidden features and applied the skew symmetric function separately on the 2 blocks generated from the mapped features.

**Evaluation Metrics:** We report three types of plots for each setup: *Bar plots:* Global feature importance scores (averaged absolute attribution values across test samples) for each method. *Spearman and Kendall Tau rank correlations* between each method's global feature importance scores and the ground truth (or baseline Integrated Gradients) are computed. *Linearity plots:* Scatter-

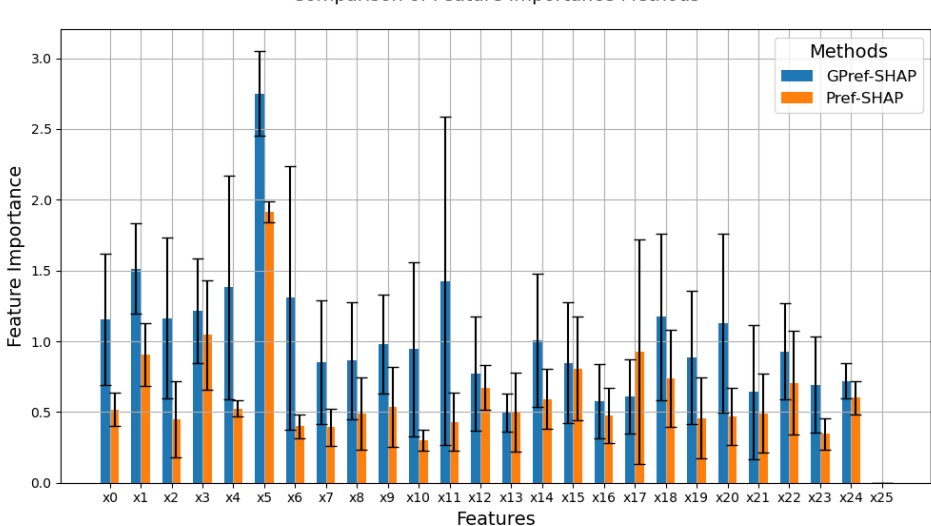

Figure 6: Pokemon dataset with 25 given features ['HP', 'Attack', 'Defense', 'Sp. Atk', 'Sp. Def', 'Speed', 'Legendary', 'Bug', 'Dark', 'Dragon', 'Electric', 'Fairy', 'Fighting', 'Fire', 'Flying', 'Ghost', 'Grass', 'Ground', 'Ice', 'Normal', 'Poison', 'Psychic', 'Rock', 'Steel', 'Water'], 'Speed'($x_5$) is the most important feature in both methods.

plots comparing predicted Pref-SHAP attributions to the sum of block-wise learned component attributions(Block-Original, Generalized Pref-SHAP), validating linearity. *RMSE curves:* Test error convergence (RMSE) across optimization iterations in the KRR model. As the Falkon library used in Pref-SHAP uses a gradient descent based hyperparameter optimization, hence for each iteration, the plot represents RMSE value between the original skew-symmetric labels and the learned labels for each algorithm. *Sanity check:* In cases where only a subset of features is used to generate the label, we verify that inactive features are assigned near-zero importance. The corresponding plots are provided in the appendix D.1.

**Observations:** Generalized Pref-SHAP consistently produces more meaningful and interpretable global feature importance scores than Pref-SHAP, particularly when the true feature importance is sparse or block-structured. Linearity plots show that both Pref-SHAP and our method yield highly correlated local feature attributions, verifying the block-decomposability of attributions in practice. RMSE plots show that Generalized Pref-SHAP models achieve comparable or better function approximation performance relative to Pref-SHAP and Block-Original. Overall, the experiments demonstrate that Generalized Pref-SHAP is more aligned with ground-truth structure in synthetic data and offers improved interpretability while maintaining the theoretical properties of the original method.

## 8 CONCLUSION

In this work, we introduced *Generalized Pref-SHAP*, a principled extension of Pref-SHAP designed to provide faithful feature attributions for pairwise preference models that involve rich, nonlinear feature mappings. We analyzed the theoretical properties of Pref-SHAP and showed that under independence or blockwise independence, Pref-SHAP decomposes additively over feature blocks, highlighting its interpretability in structured models. Our synthetic experiments demonstrate that Generalized Pref-SHAP recovers global feature importance more accurately, aligns better with ground-truth structure, and satisfies desirable attribution properties such as linearity and sparsity in inactive features whereas the real data experiment shows that *Generalized Pref-SHAP* has comparable performance with Pref-SHAP while explaining the important features for the paiwise preferences.

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

# A  APPENDIX

**Detailed Explanation of the Algorithm Steps:**

1. **Learning the Feature Mapping $\phi$:** The algorithm begins by learning a feature mapping $\phi = (\phi_1, \phi_2, \ldots, \phi_k)$ via a neural network architecture (illustrated in Figure 2). This network takes as input a pair of items $(x_l, x_r)$ with original features in $\mathbb{R}^d$, and outputs their transformed features. The learned mapping $\phi$ is structured to reflect the block-wise decomposition, where each pair $(\phi_{2i-1}, \phi_{2i})$ corresponds to a canonical $2 \times 2$ skew-symmetric block. This decomposition explicitly encodes the skew-symmetric structure in the feature space.

2. **Constructing Blockwise Datasets:** After learning the feature mappings, the dataset is partitioned into $k/2$ separate block datasets, each corresponding to the pairwise interaction between the mapped features $\phi_{2i-1}$ and $\phi_{2i}$. These datasets isolate the contribution of each skew-symmetric block to the overall preference prediction.

3. **Fitting Kernel Ridge Regression (KRR) Models:** For each block dataset, Kernel Ridge Regression is applied to learn a component function that models the preference contribution from that specific block. KRR allows flexible, nonparametric fitting that captures the potentially complex relationships within each block.

4. **Modeling the Residual:** The sum of predictions from all blockwise KRR models provides a partial reconstruction of the original preference function. This approximation may not capture all nuances of the data, particularly nonlinearities or interactions not aligned with the learned blocks. We compute the residual as the difference between the true labels $Y$ and the aggregated predictions from the blockwise models, thereby capturing information not explained by the canonical blocks. A separate KRR model is then fit to this residual dataset, enabling the method to recover additional signal beyond what is captured by the blockwise decomposition.

5. **Computing Pref-SHAP Values:** Pref-SHAP values are computed individually for each of the $k/2$ block component models and the residual model. This yields a matrix of attributions of size $k \times (k/2 + 1)$, where each column corresponds to a block (or residual), and each row corresponds to one of the $k$ mapped features. These attributions represent the contribution of each mapped feature within its block to the preference prediction.

6. **Aggregating Feature Attributions:** Finally, to obtain the attributions for the original input features, the algorithm sums the Pref-SHAP values across all blocks and the residual for each original feature. This aggregation consolidates the blockwise contributions and residual effects into a single attribution score per feature, reflecting the overall importance of each input feature in driving the preference decisions.

# B  PROPOSITION 1

*Proof.* We prove the proposition in three steps: base case $d = 4$, generalization to arbitrary even $d$, and failure of decomposition under correlation.

**Step 1: Base case** ($d = 4$)

Partition features into two blocks:

$$B_1 = \{1, 2\}, \quad B_2 = \{3, 4\}.$$

By assumption, the preference function decomposes additively:

$$f(u, v) = f_1(u_{B_1}, v_{B_1}) + f_2(u_{B_2}, v_{B_2}).$$

For feature $i = 1$, the Pref-SHAP value is

$$\Phi_1 = \sum_{S \subseteq \{2,3,4\}} w(S) \big[ \nu(S \cup \{1\}) - \nu(S) \big], \quad w(S) := \frac{|S|!(3 - |S|)!}{4!}.$$

By linearity of expectation,

$$\nu(S) = \mathbb{E}\big[ f(u, v) \mid (u_k, v_k)_{k \in S} \big] = \mathbb{E}[f_1 \mid S] + \mathbb{E}[f_2 \mid S].$$

Blockwise independence implies that conditioning factorizes:

$$\mathbb{E}[f_j \mid S] = \mathbb{E}[f_j \mid S \cap B_j], \quad j = 1, 2.$$

Since $1 \in B_1$, the marginal contribution satisfies

$$\nu(S \cup \{1\}) - \nu(S) = \big[ \nu_1((S \cup \{1\}) \cap B_1) - \nu_1(S \cap B_1) \big] + 0,$$

because $(S \cup \{1\}) \cap B_2 = S \cap B_2$ and $f_2$ does not change.

Decompose $S$ as $S = S_1 \cup S_2$, where

$$S_1 = S \cap B_1 \subseteq \{2\}, \quad S_2 = S \cap B_2 \subseteq \{3, 4\}.$$

Then,

$$\Phi_1 = \sum_{S_1 \subseteq \{2\}} \sum_{S_2 \subseteq \{3,4\}} w(S_1 \cup S_2) \big[ \nu_1(S_1 \cup \{1\}) - \nu_1(S_1) \big].$$

The marginal contribution depends only on $S_1$, so

$$\Phi_1 = \sum_{S_1 \subseteq \{2\}} \big[ \nu_1(S_1 \cup \{1\}) - \nu_1(S_1) \big] \underbrace{\sum_{S_2 \subseteq \{3,4\}} w(S_1 \cup S_2)}_{\text{weight sum}}.$$

By the combinatorial properties of Shapley weights, the inner sum over $S_2$ equals the Shapley weight on block $B_1$:

$$\sum_{S_2 \subseteq \{3,4\}} w(S_1 \cup S_2) = \frac{|S_1|!(|B_1| - 1 - |S_1|)!}{|B_1|!} = \frac{|S_1|!(1 - |S_1|)!}{2!}.$$

Thus,

$$\Phi_1 = \sum_{S_1 \subseteq \{2\}} \frac{|S_1|!(1 - |S_1|)!}{2!} \big[ \nu_1(S_1 \cup \{1\}) - \nu_1(S_1) \big].$$

Applying the same argument to feature 2, and then summing $\Phi_1 + \Phi_2$ yields

$$\Phi_1 + \Phi_2 = f_1(u_{B_1}, v_{B_1}).$$

Similarly, $\Phi_3 + \Phi_4 = f_2(u_{B_2}, v_{B_2})$.

**Step 2: Generalization to arbitrary even $d$**

For $i \in B_j$, decompose any subset $S \subseteq [d] \setminus \{i\}$ as

$$S = S_j \cup S_{-j}, \quad S_j \subseteq B_j \setminus \{i\}, \quad S_{-j} \subseteq [d] \setminus (B_j \cup \{i\}).$$

Using linearity,

$$\nu(S) = \sum_{m=1}^{d/2} \nu_m(S \cap B_m).$$

By blockwise independence,

$$\nu_m(S \cap B_m) = \nu_m(S_j \cap B_m) \quad \text{if } m = j,$$

and conditioning on $S_{-j}$ does not affect $\nu_j$.

Hence,

$$\nu(S \cup \{i\}) - \nu(S) = \nu_j(S_j \cup \{i\}) - \nu_j(S_j).$$

Then,

$$\Phi_i = \sum_{S_j \subseteq B_j \setminus \{i\}} \sum_{S_{-j}} w(S_j \cup S_{-j}) \big[ \nu_j(S_j \cup \{i\}) - \nu_j(S_j) \big].$$

Marginal contributions depend only on $S_j$, so summing weights over $S_{-j}$ gives

$$\sum_{S_{-j}} w(S_j \cup S_{-j}) = \frac{|S_j|!(|B_j| - 1 - |S_j|)!}{|B_j|!},$$

the Shapley weight inside block $B_j$.

Therefore,

$$\Phi_i = \sum_{S_j \subseteq B_j \setminus \{i\}} \frac{|S_j|!(|B_j| - 1 - |S_j|)!}{|B_j|!} \big[ \nu_j(S_j \cup \{i\}) - \nu_j(S_j) \big].$$

Summing over all $i \in B_j$,

$$\sum_{i \in B_j} \Phi_i = f_j(u_{B_j}, v_{B_j}).$$

**Special Case: Full Independence.** When all features are mutually independent (i.e., each feature forms its own block), the block decomposition reduces to:

$$f(u, v) = \sum_{j=1}^{d} f_j(u_j, v_j),$$

with $B_j = \{j\}$. Since each block now contains only one feature, the Pref-SHAP attribution for feature $i$ becomes:

$$\Phi_i = \nu_i(\{i\}) - \nu_i(\emptyset) = f_i(u_i, v_i),$$

because the Shapley value over a singleton block reduces to the full contribution of that feature. Thus, Pref-SHAP values are fully local and additive over individual features, and the global attribution decomposes as:

$$\sum_{i=1}^{d} \Phi_i = f(u, v).$$

This recovers the case of *full additivity* and *local interpretability* under complete feature independence.

**Step 3: Necessity and failure under correlation**

If features are correlated across blocks, the conditional expectation does not factorize:

$$\mathbb{E}[f_j \mid S] \neq \mathbb{E}[f_j \mid S \cap B_j].$$

Thus,

$$\nu(S \cup \{i\}) - \nu(S)$$

depends on conditioning on features outside $B_j$.

This breaks the factorization of Shapley weights and the blockwise decomposition of Pref-SHAP values fails.

**Remark:**
The Shapley value's symmetry axiom ensures that features with identical marginal contributions receive equal attribution, enabling cancellation of terms corresponding to conditioning on outside blocks when independence holds.

Linearity of expectation and combinatorial properties of weights guarantee that summation over subsets outside a block sums to one, allowing the reduction of sums to blockwise Shapley values.

This combination of linearity, independence, and symmetry underpins the blockwise decomposition of Pref-SHAP values.

$\square$

## C STUDY OF PREF-SHAP PROPERTIES W.R.T THE CANONICAL FORM

### C.1 EFFECT OF CONSTANT FEATURES ON PREF-SHAP

- Consider a skew-symmetric function with features $u = [u_1, u_2]$, $v = [v_1, v_2]$, where each item is drawn from a bivariate Gaussian distribution with zero mean:
$$Z_1 = [X_1, Y_1]^\top, \quad Z_2 = [X_2, Y_2]^\top, \quad Z_1, Z_2 \sim \mathcal{N}(0, \Sigma).$$
From (10), as $\mathbb{E}[X_i] = \mathbb{E}[Y_i]$, $\forall i$, we find $\Phi_1 \neq \Phi_2$ in general. Even if $\mathbb{E}[X_i] = \mathbb{E}[X_j]$ $\forall i, j$, $\Phi_1$ and $\Phi_2$ differ unless additional constraints hold.

- Even when one feature is kept constant (e.g., $u_2 = v_2$), it may still receive a nonzero Shapley value. For example:
$$\Phi_1 = \frac{1}{2}(u_1 - v_1)(u_2 + \mathbb{E}[X_1]), \Phi_2 = \frac{1}{2}(u_1 - v_1)(u_2 - \mathbb{E}[X_1]).$$
If $u_2 = v_2 = \mathbb{E}[X_1]$, then $\Phi_2 = 0$, but $\Phi_1 = f(u, v)$.

- If $\mathbb{E}[X_1] = 0$, then $\Phi_1 = \Phi_2 = \frac{1}{2}(u_1 - v_1)u_2$. Thus, the Shapley values are equal, and this equality is due to the *symmetry axiom*. The symmetry axiom states that two features should receive equal Shapley values if they contribute equally across all coalitions. For the two-feature case, if $v(\{1\}) = v(\{2\}) \Rightarrow \Phi_1 = \Phi_2$, even when one of the features is held constant.

- To explore this further, define:
$$v(\{1\}) = \frac{1}{2}\left(u_1 \mathbb{E}[Y_2 \mid Y_1 = v_1] - v_1 \mathbb{E}[X_2 \mid X_1 = u_1]\right),$$
$$v(\{2\}) = \frac{1}{2}\left(v_2 \mathbb{E}[X_1 \mid X_2 = u_2] - u_2 \mathbb{E}[Y_1 \mid Y_2 = v_2]\right).$$

If the features are independent, the conditional expectations reduce to marginals, and equality $v(\{1\}) = v(\{2\})$ implies:
$$(u_1 - v_1)\mathbb{E}[X_2] = (v_2 - u_2)\mathbb{E}[X_1] \tag{13}$$

If $\mathbb{E}[X_1] = \mathbb{E}[X_2] = 0$, then $\Phi_1 = \Phi_2$ always. But, if $\mathbb{E}[X_1] = \mathbb{E}[X_2] \neq 0$, then
$$\Phi_1 = \Phi_2 \implies u_1 - v_1 = v_2 - u_2$$
$$\implies u_1 + u_2 = v_1 + v_2 \tag{14}$$

- This may appear counterintuitive: the constant feature may get equal attribution even though it does not vary. This is not an artifact of the conditional expectation but a consequence of the symmetry axiom. Even if a feature is constant or non-informative in terms of variation, Pref-SHAP may still assign it equal attribution simply due to how it appears in the function and due to symmetry, unless the distribution is shifted. This suggests, Pref-SHAP attributions are not just about feature importance in terms of variance or marginal effect, but also about how the feature interacts structurally in the model and in the coalitional expectations.

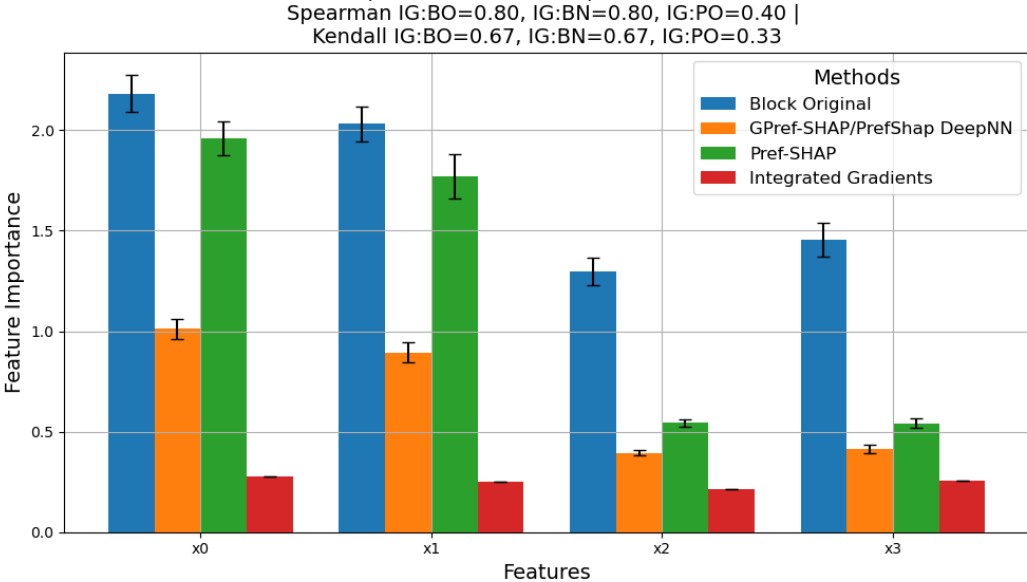

Figure 7: $x_0^2, x_1^2, x_2^2, x_3^2$

- The model function $f(u,v) = u^\top A v$ involves pairwise products of features (e.g., $u_1 v_2 - u_2 v_1$). Thus, the contribution of one feature depends on its interaction with the other. A constant feature interacting with a varying one may still result in a nonzero attribution.

- This motivates the use of *Pairwise Interaction Shapley values*[Sundararajan et al. (2020),Fumagalli et al. (2024)], which quantify pairwise contributions directly. Interaction Shapley methods can attribute the output more intuitively in models where the output depends primarily on feature interactions, as is the case in skew-symmetric functions. These interactions are model-driven, and statistical correlation (e.g., in the Gaussian setting) further modulates their effect when conditional value functions are used.

## D  ADDITIONAL SYNTHETIC DATA

We also conduct experiments using synthetic data consisting of $n = 100$ items, where each item's features are sampled from a 4-dimensional Gaussian distribution with zero mean and a covariance matrix whose diagonal entries are 1 and off-diagonal entries are 0.7:

$$x^{(i)} \sim \mathcal{N}(0, \Sigma), \quad \text{for } i = 1, \dots, 100,$$

where $\Sigma_{jj} = 1$ and $\Sigma_{jk} = 0.7$ for $j \neq k$.

Figures 14, 15 and 16 are the synthetic experiments conducted on such highly correlated data. If we compare figures 16 and 17, we can see that when the features are highly correlated, the dummy/inactive feature $x_3$ gets more attribution than the case when the features are independent.

### D.1  SANITY CHECK

Figures 17,18,19,20 are some experiments used for sanity check. In figure 17, the fourth/last feature is not used for generating mapped features and hence the skew-symmetric function. In figures 18, 19 and 20 items are generated using a 6-dimensional Gaussian distribution with mean zero and identity covariance: $x_i \sim \mathcal{N}(0, \mathbf{I})$ for $i = 1, \dots, 100$. , but only the first 4 features are used for feature mapping and label generation, so the last two features act like dummy ones. We can observe from these plots that the dummy features get zero attribution in case of *Integrated Gradients* whereas Block Original and Generalized Pref-SHAP have nearly-zero or the lowest attribution for

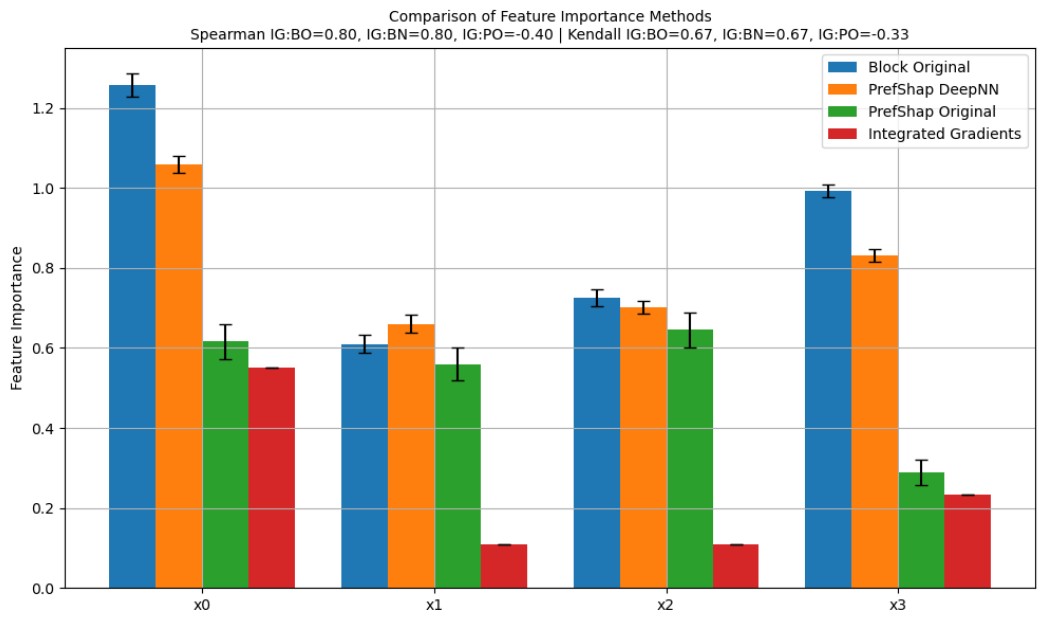

Figure 8: $x_0^2, x_1 x_2, \sin(x_3), \cos(x_0)$

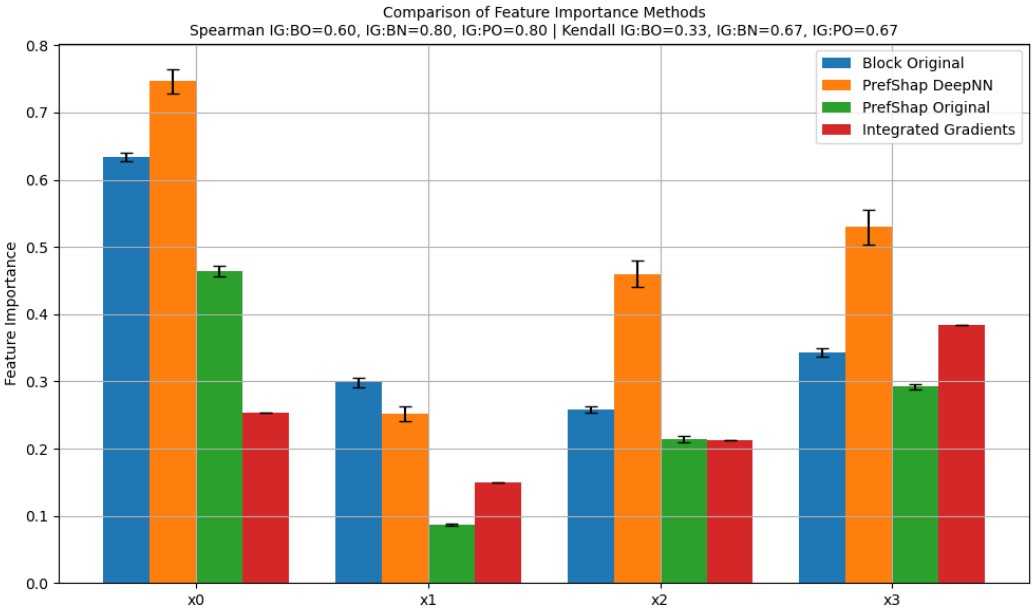

Figure 9: $sin(x_0), cos(x_1), sin(2x_2), cos(2x_3)$

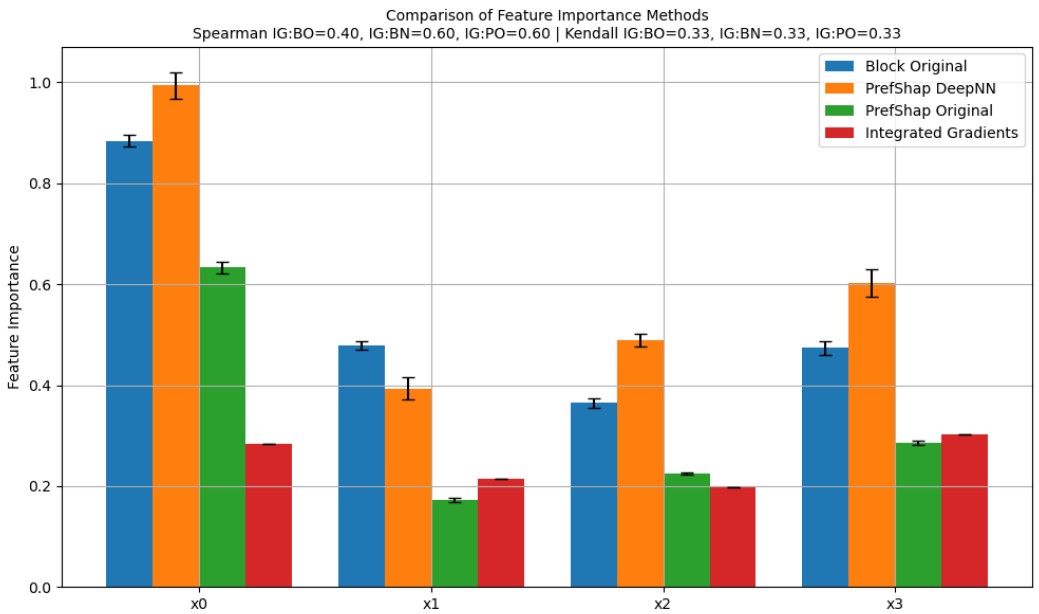

Figure 10: $sin(x_0), cos(x_1), sin(2x_2), cos(2x_3), sin(x_0 + x_1), cos(x_2 - x_3)$

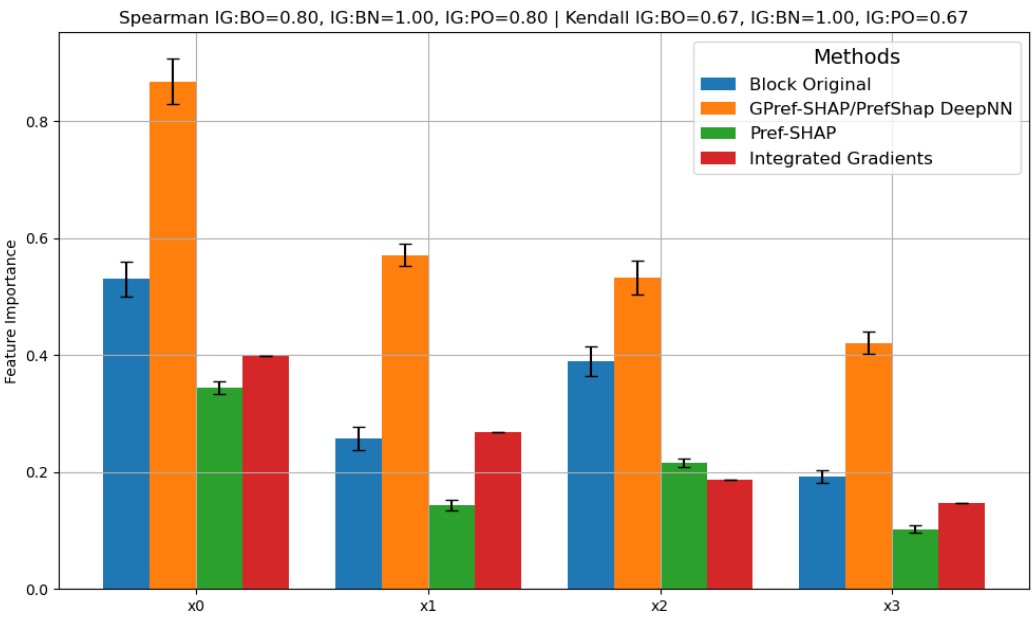

Figure 11: ReLU network(4 input features, 4 mapped features, 8 hidden layers, 16 nodes in each layer)

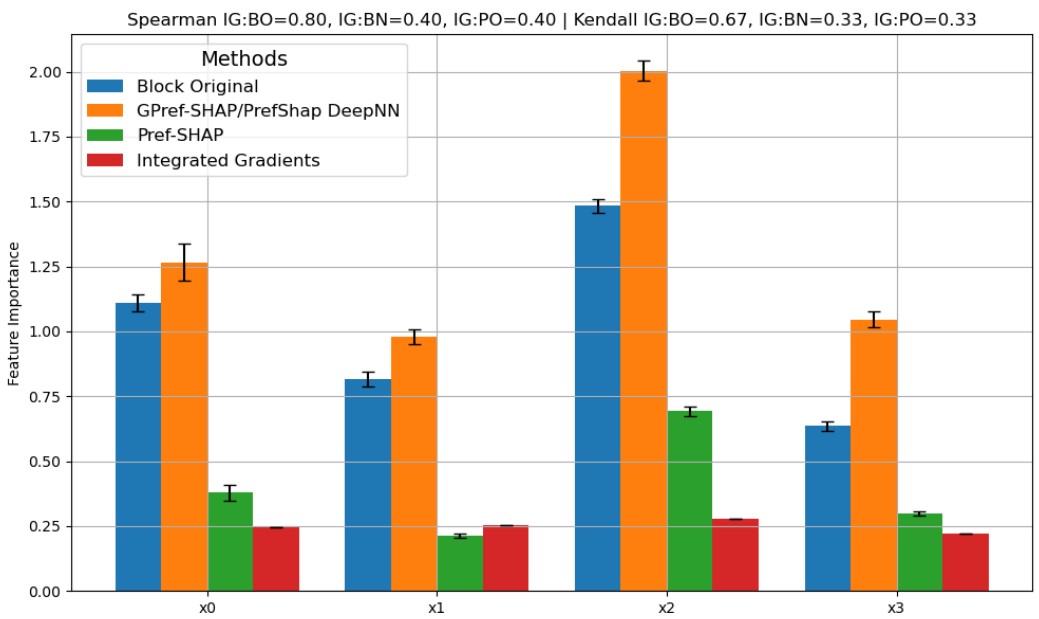

Figure 12: ReLU network(4 input features, 6 mapped features, 8 hidden layers, 8 nodes in each layer)

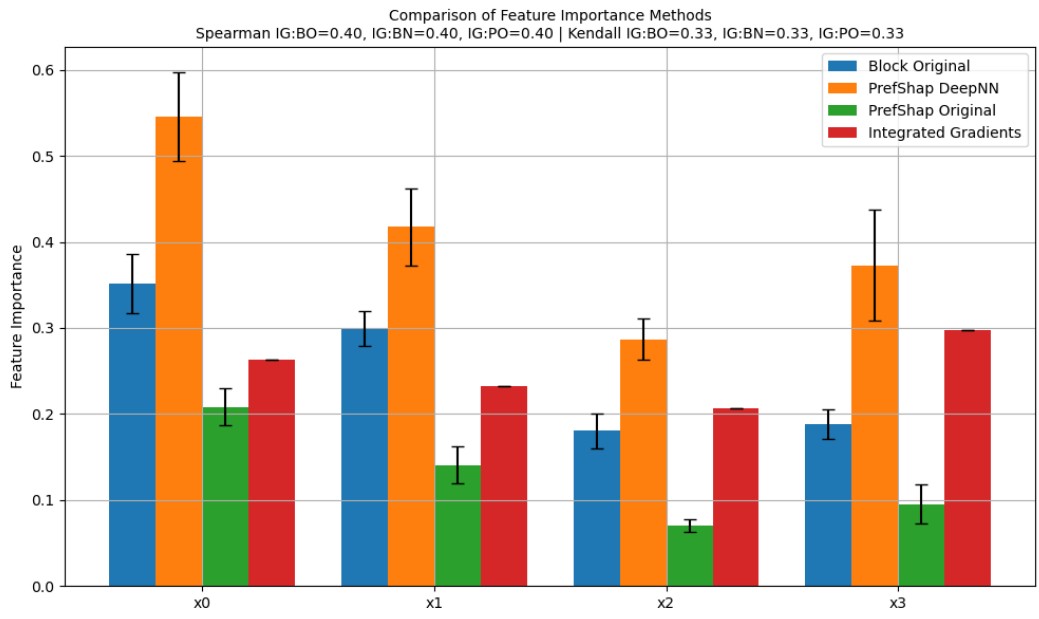

Figure 13: ReLU network(4 input features, 4 mapped features, 16 hidden layers, 16 nodes in each layer)

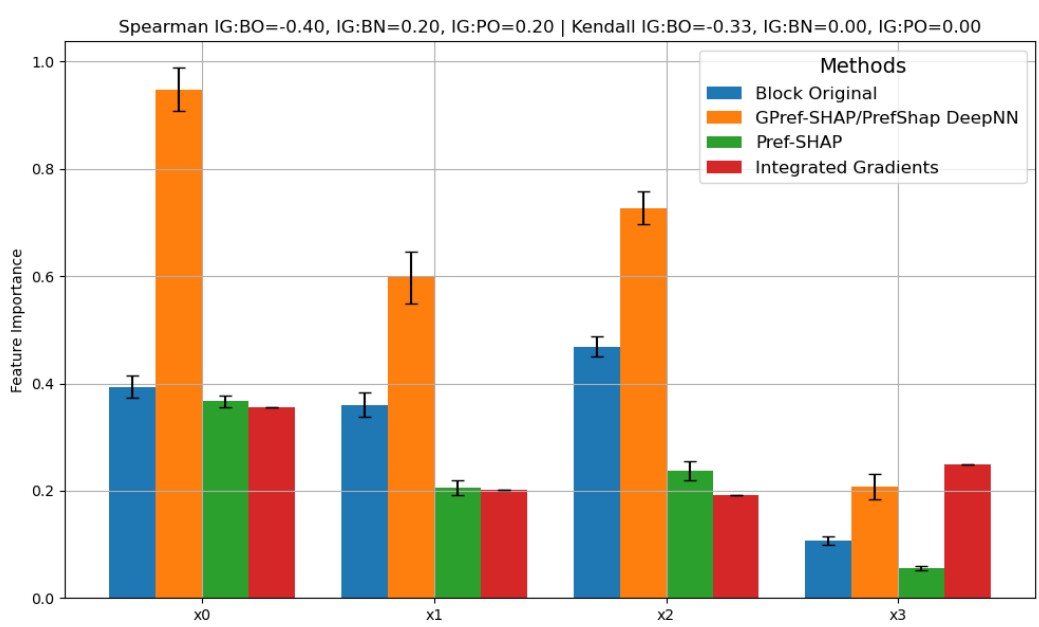

Figure 14: ReLU network(4 input features, 4 mapped features, 8 hidden layers, 16 nodes in each layer) for highly correlated data

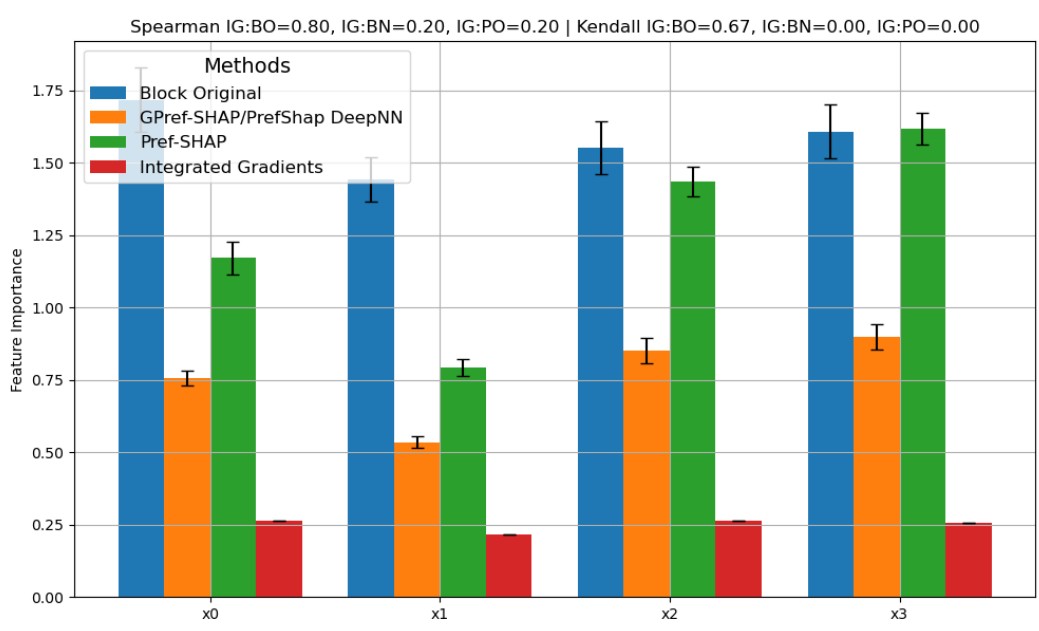

Figure 15: $x_0^2, x_1^2, x_2^2, x_3^2$ for highly correlated data

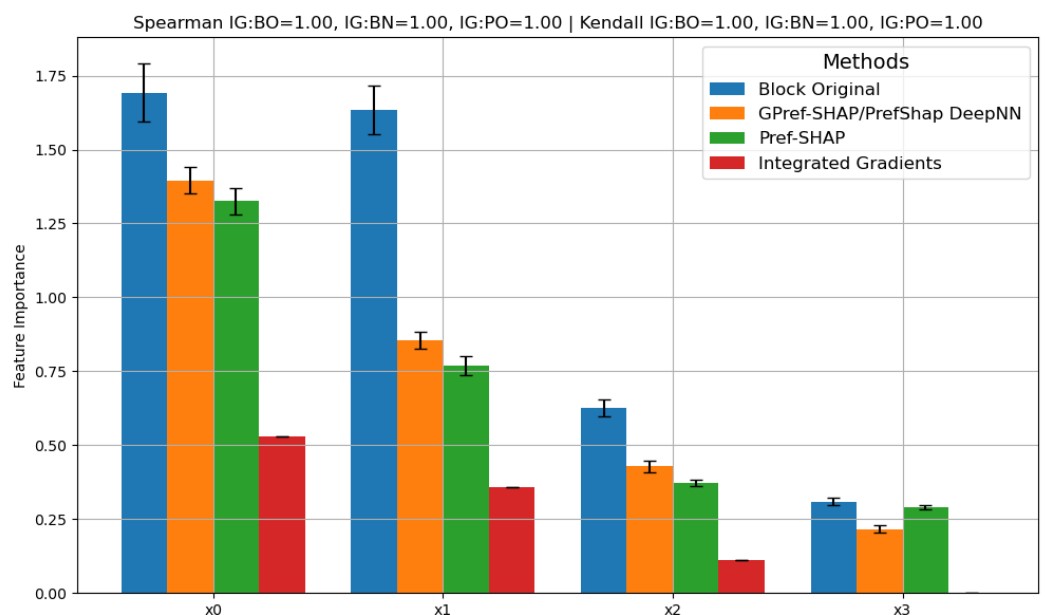

Figure 16: $x_0^2, x_1^2, x_0x_1, x_0x_2$ for highly correlated data

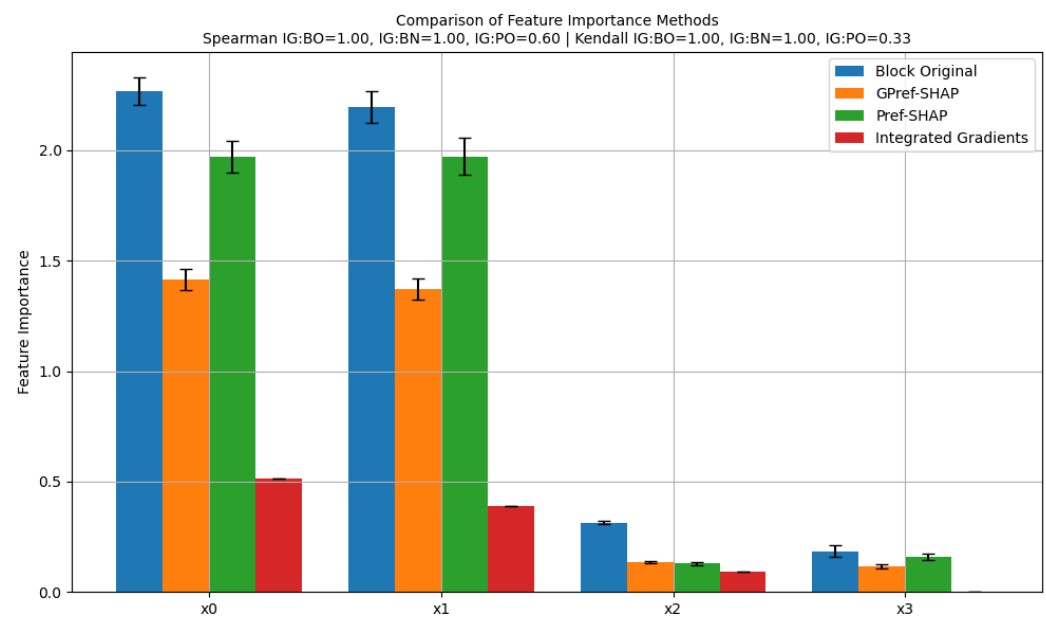

Figure 17: $x_0^2, x_1^2, x_0x_1, x_0x_2$ with $x_3$ as the inactive feature

such features in all of them. But in figure 18, Pref-SHAP has more attribution for the dummy feature $x_5$ than $x_0$ which is an active feature in the function generation. Also, in figure 17, Pref-SHAP has more feature attribution for $x_3$ than that of $x_2$.

## E    TEST RMSE AND LINEARITY SCATTER PLOTS FOR THE SYNTHETIC DATA

Figures 21,22,23,24 represent the plots for test RMSE and scatter plots for the synthetic datasets.

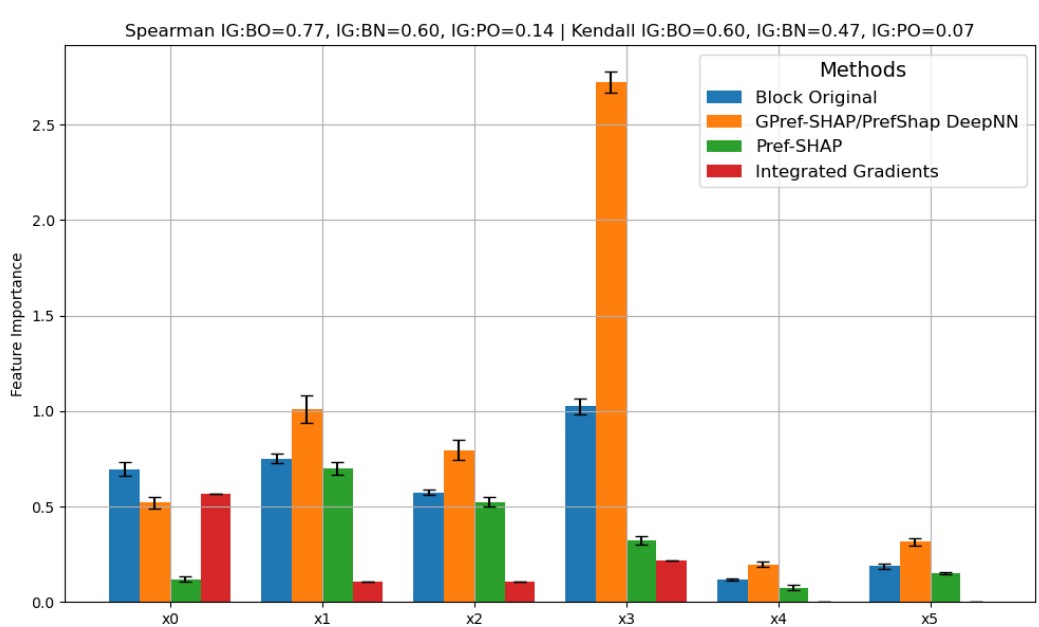

Figure 18: $x_0^2, x_1 x_2, \sin(x_3), \cos(x_0)$ with 2 inactive input features

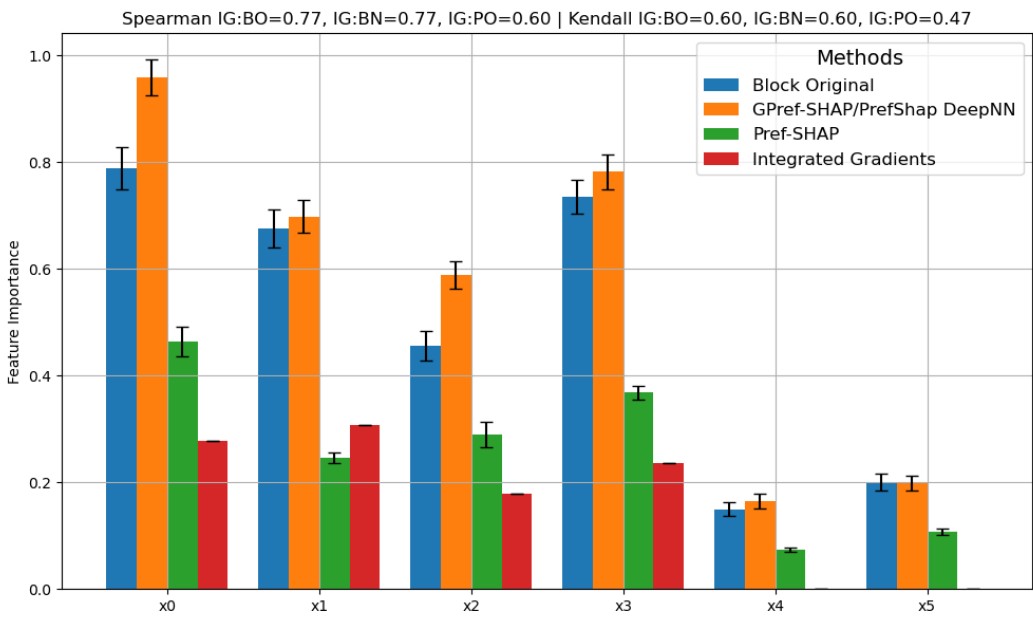

Figure 19: ReLU network(4 active + 2 inactive input features, 6 mapped features, 4 hidden layers, 16 nodes in each layer)

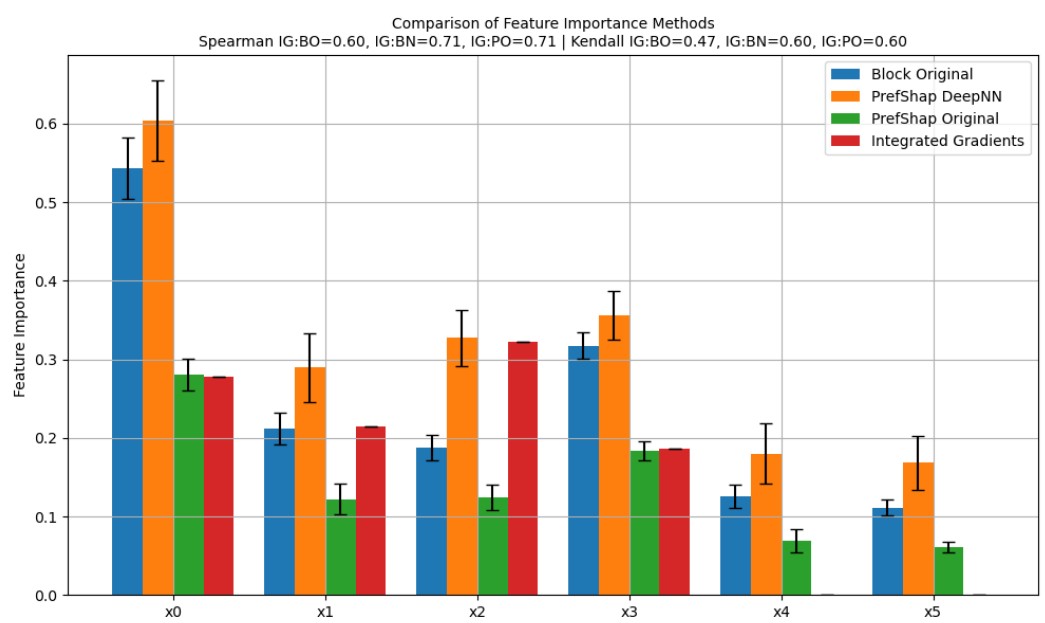

Figure 20: ReLU network(4 active + 2 inactive input features, 4 mapped features, 16 hidden layers, 16 nodes in each layer)

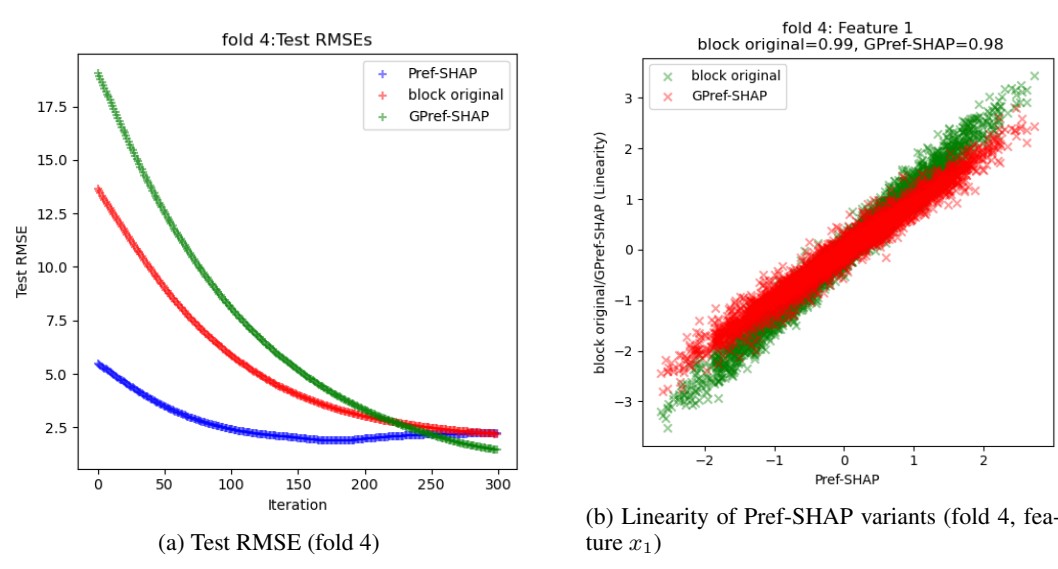

(a) Test RMSE (fold 4)

(b) Linearity of Pref-SHAP variants (fold 4, feature $x_1$)

Figure 21: $x_0^2, x_1 x_2, \sin(x_3), \cos(x_0), \sin(x_1 + x_2), x_3^2$

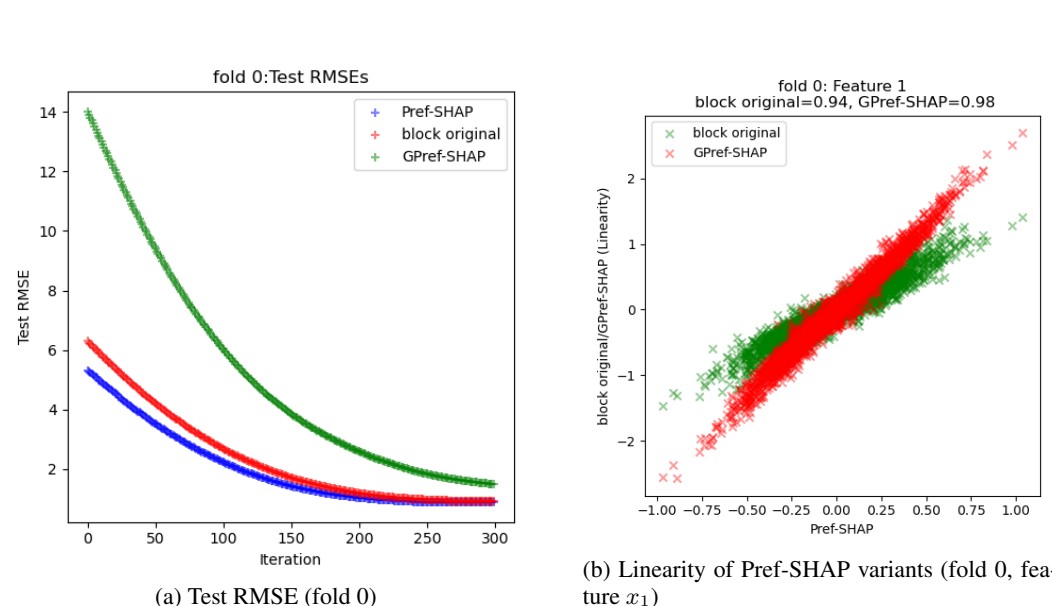

(a) Test RMSE (fold 0)

(b) Linearity of Pref-SHAP variants (fold 0, feature $x_1$)

Figure 22: ReLU network(4 input features, 4 mapped features, 8 hidden layers, 16 nodes in each layer) with highly correlated features

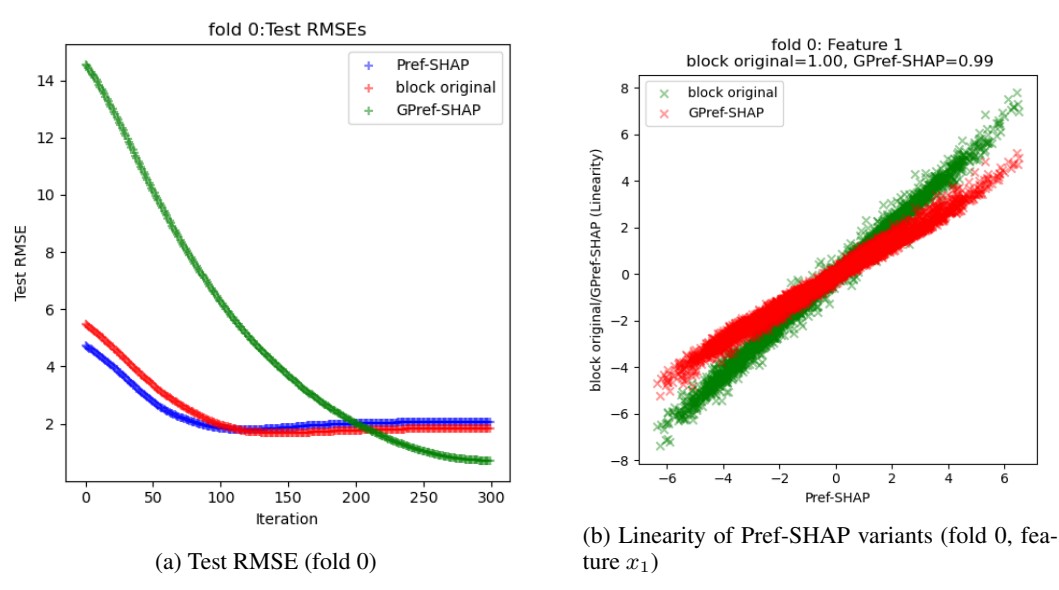

(a) Test RMSE (fold 0)

(b) Linearity of Pref-SHAP variants (fold 0, feature $x_1$)

Figure 23: $x_0^2, x_1^2, x_0x_1, x_0x_2$ with independent features

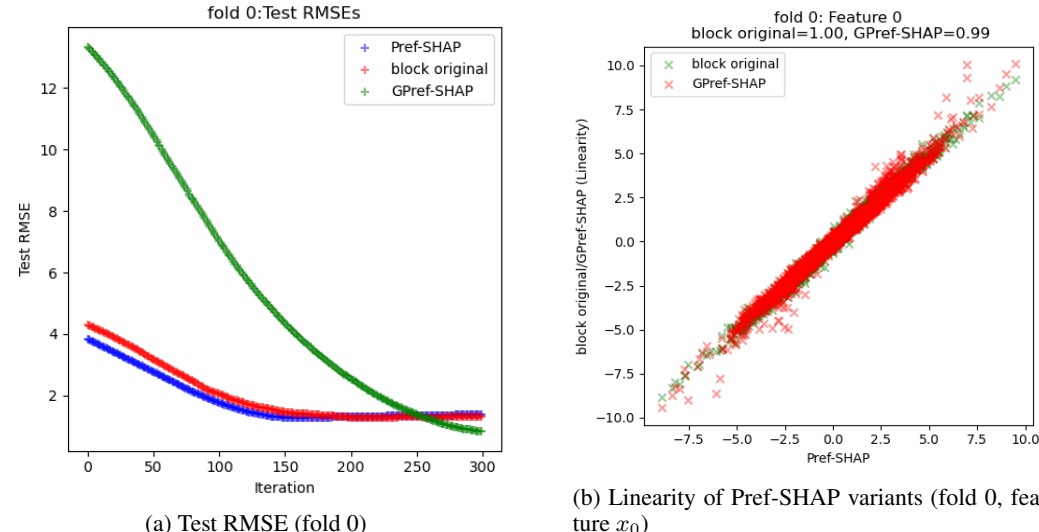

(a) Test RMSE (fold 0)

(b) Linearity of Pref-SHAP variants (fold 0, feature $x_0$)

Figure 24: $x_0^2, x_1^2, x_0x_1, x_0x_2$ with highly correlated features

## F USE OF LLMS:

Chatgpt has been used to polish the writing of certain parts in the paper.

