# OpenReview forum: "Generalized Pref-SHAP to Explain Preference Functions"
_ICLR.cc/2026/Conference — ICLR 2026 Conference Withdrawn Submission_

### Official Review · Reviewer_8dmy · 2025-10-17

**Soundness:** 2
**Presentation:** 1
**Contribution:** 2
**Rating:** 2
**Confidence:** 5

**Summary:**

This paper introduces Generalized Pref-SHAP (GPref-SHAP), an extension of the Pref-SHAP framework designed to explain skew-symmetric preference function. The authors identify a key limitation in the original Pref-SHAP: it fails to respect the inherent "block structure" of certain preference models when input features are statistically correlated. This can cause feature attributions to "leak" across conceptually separate feature blocks, potentially producing misleading explanations.

To address this, GPref-SHAP proposes an explanation-aware modeling approach. The method uses a neural network to explicitly learn a feature mapping, $\phi$, from the original inputs. This learned representation is then fed into a fixed architecture that computes the preference score as a sum of decomposed, block-wise interactions

By enforcing this functional decomposition, the model is claimed to become interpretable by design, ensuring the resulting explanations align with its structure. Experiments on synthetic data claim that GPref-SHAP recovers ground-truth feature importance more accurately than the original Pref-SHAP and performs better on sanity checks involving inactive features.

**Strengths:**

The paper makes a good point on the independence assumption Pref-SHAP makes and how certain pathologies arise during computation of Shapley values when this is violated. The arguments for this is somewhat well presented.

**Weaknesses:**

1. The proposed method immediately proposes neural network and the additional integrated gradient approach to get interpretable features from the block format. What are the computational overhead for this, Shapley values are generally expensive to compute so adding complexity and overhead must be done with great care. It would have been more convincing to introduce an example where Pref-SHAP explicitly breaks due to this correlation, and to quantify at what degree of multicollinearity Pref-SHAP breaks.
2. In one of the experiments the proposed method seems to hamper predictive performance, how does improved explanations trade-off against predictive performance - i.e. when is it better to have slightly wrong explanations for a model that predicts very well or accurate explanations for a model that predicts wrongly.
3. The exposition of the paper needs work, the plots are not very well formatted and unnecessarily big.
4. Some more real-life experiments would strengthen the paper.
5. In the Pokemon experiment, Pref-SHAP also finds speed to be the most important feature similar to GPref-SHAP. Is there case here where GPref-SHAP finds insights Pref-SHAP is unable to derive?

**Questions:**

1. When exactly does Pref-SHAP break? Can you provide a minimal empirical example that demonstrates using data exhibiting multicollinearity?
2. Do we really need a neural network to satisfy the block requirement? Can't we just do 2-SLS or remove highly correlated features and see if things improve?

---

> ### Author Response · Authors · 2025-12-04
> **Response to Reviewer 8dmy**
>
> We thank the reviewer for the detailed feedback.
>
> The block pattern in our paper is a theoretical construct used to analyze the decomposition and interpretability of Pref-SHAP under independence (Proposition 1). The purpose of the neural network is to simply learn a general nonlinear feature mapping $\phi$ using the inherent block structure of the preference function $f(u,v) = \langle \phi(u), A \phi(v) \rangle$, not to satisfy any block requirement.
>
> Regarding the computational overhead, we agree that Generalized Pref-SHAP introduces additional computation. The method first learns a small neural representation $\phi$ (two hidden layers, ≤128 units) and then fits multiple FALKON-based KRR models, one per each canonical block of mapped features. Although each KRR still receives the full $d$-dimensional raw features as input (same as in Pref-SHAP), the computational complexity of FALKON is $O(N\sqrt N)$ and is independent of the input dimensionality. Therefore the total cost simply scales linearly with the number of blocks ($k/2$), and remains practical in all experiments. This overhead is the necessary cost for obtaining faithful block-consistent attributions when nonlinear learned representations are present, settings where Pref-SHAP’s raw-feature surrogate is no longer adequate.
>
> Our method does not alter the predictive model used for inference. The RMSE curves reflect only the accuracy of the surrogate functions used for computing Shapley values (a single KRR for Pref-SHAP vs. block-wise KRR for Generalized Pref-SHAP). The block-wise surrogate used by Generalized Pref-SHAP is intentionally more structured and therefore less expressive, which explains the higher RMSE. This does not imply a loss in predictive performance, it only reflects the structural constraints needed to obtain faithful block-consistent explanations in nonlinear representation spaces. Generalized Pref-SHAP provides explanations that are faithful to the model’s actual nonlinear representation structure, whereas Pref-SHAP explains a linearized surrogate when such structure exists. When the model is close to linear, the two explanations coincide. When the model relies on nonlinear representations, Generalized Pref-SHAP produces more reliable attributions without affecting the model’s predictions.
>
> Please check the updated rebuttal version of the paper. Most of the presentation issues have been fixed.
>
> We agree that some more real-life experiments would strengthen the paper. But as mentioned in the paper, we have conducted experiments mainly on carefully generated synthetic data because the motivation behind Generalized Pref-SHAP is rooted in the design of feature mappings $\phi$, and real-world dueling datasets rarely provide a ground truth for global feature importance. Synthetic datasets are the only environment where interpretability can be objectively evaluated. We therefore include one real-world dataset strictly to show practical applicability, not to measure ground truth.
>
> On the Pokémon dataset, Pref-SHAP and Generalized Pref-SHAP indeed produce very similar attributions. This empirical agreement indicates that, for this particular dataset, the learned preference model behaves approximately linearly in the raw features. So Pref-SHAP’s linear surrogate is already a good approximation of the learned model. In such cases, a nonlinear representation does not introduce additional structure, and both methods naturally highlight the same dominant features (e.g., Speed). The purpose of this experiment is therefore not to show divergence between the two methods, but to verify that Generalized Pref-SHAP reduces to Pref-SHAP when the linearity assumptions of Pref-SHAP approximately hold. This is an important sanity check: a generalization of a method should match the original in regimes where the original is valid. The benefits of Generalized Pref-SHAP appear in the nonlinear settings, shown in our synthetic experiments, where Pref-SHAP exhibits leakage and block-inconsistency that Generalized Pref-SHAP corrects.

---

### Official Review · Reviewer_B9Cf · 2025-10-29

**Soundness:** 1
**Presentation:** 1
**Contribution:** 1
**Rating:** 0
**Confidence:** 3

**Summary:**

This paper investigates an extension to PrefSHAP by considering a more general class of preference functions. The authors provide some results on the (extended) value function in this setting (Proposition 1), which uses feature independence or a "block structure". The authors then provide an example, and study theoretical properties in a setting with two features. Moreover, an algorithm is presented, and some empirical results are presented.

**Strengths:**

- Investigating more general forms of preference learning with Shapley values could possibly be interesting. However, an efficient computation of such Shapley values would be desirable by exploiting certain properties of this novel value function. I am doubtful, if this extensions will yield such insights.

**Weaknesses:**

In my view, this paper, in its current form, should not have been submitted at any conference, and is far from most scientific standards: There are obvious formatting issues, e.g. typos, citations, exceeded margins, no clear structure, figures and descriptions are chaotic. Moreover, the contribution is very unclear. Block structures are not introduced well, the purpose of the algorithm is not clear at all. I did not understand any central part of the contribution, e.g. Section 3 discusses the Block Pattern, what should that be? I did not understand the example and its purpose. The properties examined in Section 4 with two features are very artificial, and I still did not understand the insight. The experiments use a single synthetic dataset with 4 (!) features. There might be some interesting insights in this method, but as it is being presented now, it is not understandable, and clearly not ready for being accepted at this conference.

**Questions:**

I do not think my questions can be sufficiently addressed by the authors in the rebuttal, but some were stated under "weaknesses".

---

> ### Author Response · Authors · 2025-12-04
> **Response to  Reviewer B9Cf**
>
> We would like to clarify that our paper does not introduce a "novel" or "extended" value function. The conditional value function we analyze is identical to that in Pref-SHAP, defined as the expectation of the preference function conditioned on a subset of features. Our theoretical contribution considers the structure of this value function when applied to skew-symmetric preference functions, not a redefinition of the Shapley operator itself. When $\phi(u) = u$, our framework reduces exactly to the Pref-SHAP formulation. The same value function is used in the generalized version of Pref-SHAP. Hence, our contribution lies in generalizing the function class rather than redefining the Shapley value framework.
>
> Most of the formatting/presentation issues have been corrected in the rebuttal version of the paper.
>
> The main three contributions of the paper are already clearly mentioned in the introduction part of the paper. The first contribution concerns "Block Structure Consistency" i.e., we analyze whether Pref-SHAP respects the block structure inherent in the canonical form. This reveals how well the Shapley attributions align with the inherent feature pairing in the model. This is covered in section 3. The second contribution (section 4) consists of the study of certain properties i.e., feature variance, constancy etc. of Pref-SHAP under the canonical form of the skew-symmetric function in the two-features setting. Our 3rd contribution (section 6) is the extension of Pref-SHAP to the Generalized Pref-SHAP framework that deals with the nonlinear feature representation.
>
> Block structures are introduced properly in section 2.6.
>
> The purpose of the algorithm is clearly mentioned in section 6. Detailed explanation of the algorithmic steps are described in appendix A.
>
> The properties examined in Section 4 with two features are not at all artificial. It consists of the analysis of certain distributional properties of Pref-SHAP like feature variance etc., in the two-features setting. Using the value function (definition 1, section 2.4) and the analytical form of Pref-SHAP presented in proposition 1 (section 3), we can easily write the analytical expression of Pref-SHAP w.r.t. the canonical form of the skew-symmetric preference function. It’s an analytic toy case to illustrate the properties, standard in theoretical papers.
>
> The synthetic experiments deliberately use a 4-feature setting because each canonical block in the skew-symmetric structure contains exactly two interacting features. A 4-feature model therefore contains two independent blocks, which is the minimal nontrivial case where one can observe block consistency, leakage effects, and the failure modes of Pref-SHAP versus the block-aware Generalized Pref-SHAP. Using higher-dimensional synthetic data does not introduce new behaviors, only more layers of the same block interactions, and it obscures the interpretability of the phenomena we want to isolate. Also, we have carried out several synthetic experiments w.r.t. a variety of non-linear feature mappings to observe the overall effect.

---

### Official Review · Reviewer_M585 · 2025-10-31

**Soundness:** 2
**Presentation:** 3
**Contribution:** 2
**Rating:** 6
**Confidence:** 3

**Summary:**

This paper introduces Generalized Pref‑SHAP, an extension of Pref‑SHAP for explaining pairwise preference functions with nonlinear feature mappings. It learns structured feature representations via neural networks while preserving the canonical skew‑symmetric block structure. The method improves interpretability, decomposability, and consistency across correlated features in preference learning models.

**Strengths:**

- The authors successfully preserve block‑wise interpretability in skew‑symmetric preference models.
- This paper demonstrates that the framework supports nonlinear and learned feature mappings, making it more generalizable.
- The proposed method achieves higher attribution accuracy and robustness compared with existing approaches.

**Weaknesses:**

- The proposed method requires higher computational cost due to neural network training and multiple KRR models.
- This paper relies on independence assumptions for certain theoretical guarantees, limiting universality.
- The authors provide limited real‑world evaluation, focusing primarily on synthetic datasets.

**Questions:**

- The proposed method requires higher computational cost due to neural network training and multiple KRR models.
- This paper relies on independence assumptions for certain theoretical guarantees, limiting universality.
- The authors provide limited real‑world evaluation, focusing primarily on synthetic datasets.

---

> ### Author Response · Authors · 2025-12-04
> **Response to Reviewer M585**
>
> We thank the reviewer for the detailed feedback.
>
> We agree that Generalized Pref-SHAP introduces additional computation. The method first learns a small neural representation $\phi$ (two hidden layers, ≤128 units) and then fits multiple FALKON-based KRR models, one per each canonical block of mapped features. Although each KRR still receives the full $d$-dimensional raw features as input (same as in Pref-SHAP), the computational complexity of FALKON is $O(N\sqrt N)$ and is independent of the input dimensionality. Therefore the total cost simply scales linearly with the number of blocks ($k/2$), and remains practical in all experiments. This overhead is the necessary cost for obtaining faithful block-consistent attributions when nonlinear learned representations are present, settings where Pref-SHAP’s raw-feature surrogate is no longer adequate.
>
> The paper does not rely on independence assumptions for any theoretical guarantees. Independence assumption is used only in Proposition 1 to characterize when the block-wise decomposition property holds, mirroring the original Pref-SHAP analysis. It is not an assumption required for the algorithm, or the general applicability of Generalized Pref-SHAP.
>
> As mentioned in the paper, we have conducted experiments mainly on carefully generated synthetic data because the motivation behind Generalized Pref-SHAP is rooted in the design of feature mappings  $\phi$, and real-world dueling datasets rarely provide a ground truth for global feature importance. Synthetic datasets are the only environment where interpretability can be objectively evaluated. We therefore include one real-world dataset strictly to show practical applicability, not to measure ground truth.

---

### Official Review · Reviewer_mbpX · 2025-11-01

**Soundness:** 1
**Presentation:** 1
**Contribution:** 1
**Rating:** 0
**Confidence:** 3

**Summary:**

This paper proposes extending Pref-SHAP (Hu et al., NeurIPS 2022), a method for explaining preference learning using Shapley values. Theoretical analysis gives a single proposition on the block decomposition of conditional Pref-SHAP under feature independence. Experiments with two synthetic tabular datasets compare the proposed Generalized Pref-SHAP to Pref-SHAP.

**Strengths:**

Unfortunately, it is challenging to find any.

**Weaknesses:**

This work resembles more a preliminary workshop contribution rather than a complete conference publication:
1. The motivation and significance of this research are weak. There are no impactful applications for the method. This is evident from the fact that experiments are conducted primarily with synthetic data. A single "real-world" example with a "Pokémon" dataset is shown in Appendix E, Figure 6. Furthermore, the paper does not reference any emerging applications.
2. Discussion of related literature is limited to only 13 references. Why is this research important? The introduction provides no context for studying such a method.
3. Presentation is subpar (see feedback below).

**Questions:**

1. Why is Appendix G empty?
2. Can this research be motivated by reinforcement learning from human feedback and preference learning for LLMs?

Feedback:
- All figures can be much smaller, saving space for actual research content. You should not write "Section 3.1: This discussion and the related literature are included in the appendix due to space constraint."
- The critical example with "real-world" data should be included in the main text.
- Figure 1 should be a Table. It is also too large in width.
- Equation (11) shouldn't exceed the margin.
- Why are citations written without parentheses like "In the Pref-SHAP framework Hu et al. (2022)," instead of "In the Pref-SHAP framework (Hu et al., 2022),"?
- L315: typo in "kernelChau et al. (2022a)"
- Figure 6 should list feature names next to bars, not in the caption.

In general, you can mimic the Pref-SHAP (Hu et al., NeurIPS 2022) paper to directly improve the presentation of this submission.

---

> ### Author Response · Authors · 2025-12-04
> **Response to Reviewer mbpX**
>
> Motivation and significance of our research-
>
> Modern preference models including deep dueling networks, Siamese models, RLHF etc., compute preferences in a learned representation, not in the raw feature space. Pref-SHAP explains predictions in the raw space, causing attribution leakage and misalignment when the model’s computation happens in some hidden space. Generalized Pref-SHAP fixes this by learning the true representation and performing block-structured Shapley attribution in that space, yielding faithful explanations for nonlinear preference models.
>
> We agree that some more real-life experiments would strengthen the paper. But as mentioned in the paper, we have conducted experiments mainly on carefully generated synthetic data because the motivation behind Generalized Pref-SHAP is rooted in the design of feature mappings $\phi$, and real-world dueling datasets rarely provide a ground truth for global feature importance. Synthetic datasets are the only environment where interpretability can be objectively evaluated. We therefore include one real-world dataset strictly to show practical applicability, not to measure ground truth.
>
> Most of the presentation issues have been fixed in the updated rebuttal version of the paper. Please refer to it to clarify any doubt regarding the importance/impactful applications of the research and to understand the context of studying such a method.
>
> Appendix G was not empty in the earlier version of the paper (now appendix E, in the rebuttal version). It contained the test RMSE plots and linearity scatterplots supporting the quantitative analysis for some additional synthetic datasets. We have made this clearer in the text to avoid any confusion.
>
> Yes, this research can be motivated by reinforcement learning from human feedback and preference learning for LLMs. Modern RLHF and LLM preference-learning models operate in a high-dimensional nonlinear representation space, where the true preference function is computed over deep embeddings rather than over raw features. Pref-SHAP does not apply in this regime, because it assumes a fixed linear feature space. GPref-SHAP is motivated precisely by this setting - it provides faithful, block-consistent Shapley explanations for nonlinear, representation-based preference models like those used in RLHF.

---

### Note · Authors · 2026-01-03

I have read and agree with the venue's withdrawal policy on behalf of myself and my co-authors.